# Inter-chromosomal transcription hubs shape the 3D genome architecture of African trypanosomes

Claudia Rabuffo [1,2], Markus R. Schmidt [1,2], Prateek Yadav [1,2], Pin Tong[3], Roberta Carloni[3,4], Anna Barcons-Simon [1,2], Raúl O. Cosentino[1,2], Stefan Krebs [5], Keith R. Matthews [4], Robin C. Allshire [3] & T. Nicolai Siegel [1,2] ✉

The eukaryotic nucleus exhibits a highly organized 3D genome architecture, with RNA transcription and processing confined to specific nuclear structures. While intra-chromosomal interactions, such as promoter-enhancer dynamics, are well-studied, the role of inter-chromosomal interactions remains poorly understood. Investigating these interactions in mammalian cells is challenging due to large genome sizes and the need for deep sequencing. Additionally, transcription-dependent 3D topologies in mixed cell populations further complicate analyses. To address these challenges, we used high-resolution DNA-DNA contact mapping (Micro-C) in *Trypanosoma brucei*, a parasite with continuous RNA polymerase II (RNAPII) transcription and polycistronic transcription units (PTUs). With approximately 300 transcription start sites (TSSs), this genome organization simplifies data interpretation. To minimize scaffolding artifacts, we also generated a highly contiguous phased genome assembly using ultra-long sequencing reads. Our Micro-C analysis revealed an intricate 3D genome organization. While the *T. brucei* genome displays features resembling chromosome territories, its chromosomes are arranged around polymerase-specific transcription hubs. RNAPI-transcribed genes cluster, as expected from their localization to the nucleolus. However, we also found that RNAPII TSSs form distinct inter-chromosomal transcription hubs with other RNAPII TSSs. These findings highlight the evolutionary significance of inter-chromosomal transcription hubs and provide new insights into genome organization in *T. brucei*.

To fit inside the nucleus of a cell, the long polymers of DNA that contain a cell's genetic information must be tightly packaged[1]. Research over the past few decades has shown that DNA is not randomly packaged and that the 3D genome organization inside the nucleus regulates vital cellular processes, including gene expression and DNA repair[2-4]. Chromosome conformation capture (3C)-based techniques, which measure the frequency of DNA-DNA contacts between any two DNA loci, have revealed a large number of distinct

[1]Division of Experimental Parasitology, Faculty of Veterinary Medicine, Ludwig-Maximilians-Universität München, 82152 Planegg-Martinsried, Germany. [2]Biomedical Center Munich, Division of Physiological Chemistry, Ludwig-Maximilians-Universität München, 82152 Planegg-Martinsried, Germany. [3]Wellcome Centre for Cell Biology, School of Biological Sciences, University of Edinburgh, Edinburgh EH9 3BF, United Kingdom. [4]Institute of Immunology and Infection Research, University of Edinburgh, Edinburgh EH9 3FL, United Kingdom. [5]Gene Center and Department of Biochemistry, Ludwig-Maximilians-Universität München, 81377 Munich, Germany. ✉e-mail: n.siegel@lmu.de

structures, collectively referred to as 'contact domains'. Although it has long been recognized that contact domains vary in size and function, the terminology used to describe different contact domains has changed over time. To reduce ambiguity, in our study we follow the definitions proposed by Beagan and Phillips-Cremins[5]. Here, 'chromatin domains' are defined as any triangle of enhanced contact frequency tiling the diagonal of a DNA-DNA contact matrix.

At the largest scale, eukaryotic DNA is typically organized into active and inactive regions that segregate into A and B compartments[6]. 'Compartment domains' are thus defined as chromatin domains whose boundaries coincide with inflection points in the A/B compartmentalization signals, and compartment boundaries are identified by computing the eigenvectors of the interaction matrices[7,8].

At the intermediate scale, genomes can be organized into 'topologically associating domains' (TADs). Beagan and Phillips-Cremins define TADs as contact domains whose formation is driven by cohesin-mediated loop extrusion[5,9,10]. These domains are flanked by CCCTC-binding factor (CTCF), a protein that restricts the movement of cohesin and thereby defines the boundaries of TADs[11–13]. TAD boundaries can be identified by calculating an insulation score that reflects the amount of interaction between two TADs and other contact domains.

At the finest scale, high-resolution 3C-based approaches in metazoans have revealed so-called 'chromatin loops', defined as a point of enriched contacts in Hi-C or Micro-C heat maps, which appear as a dot (a series of adjacent pixels with enhanced contact frequency relative to the local chromatin domain structure). Such loops can represent distinct interactions between enhancers and promoters that may be important for gene regulation[14–17].

Additionally, microscopy[18,19] and methods capable of capturing long-distance and multivalent interactions, such as split-pool recognition of interactions by tag extension (SPRITE)[20], have shown that specific inter-chromosomal interactions can organize around distinct nuclear bodies. For example, RNA polymerase I (RNAPI)-transcribed rDNA genes, encoded on several different chromosomes, localize within the nucleolus[21]. Although dispersed in the linear genome, tRNA genes transcribed by RNA polymerase III (RNAPIII) can form spatial clusters[22–24], which facilitate the coordination of transcription and processing of tRNAs, crucial for protein synthesis. Similarly, RNA polymerase II (RNAPII)-transcribed genes have been found to cluster in structures, sometimes referred to as transcription factories[25,26], condensates[27], pockets[28] or hubs[29], which are discrete nuclear sites where active transcription occurs. These clusters can bring together multiple genes from different regions of the genome to be transcribed simultaneously.

The organization of genes into inter-chromosomal hubs has also been reported to be important beyond coordinating transcription and to contribute to efficient RNA processing. *Trypanosoma brucei* is a highly divergent, unicellular parasite responsible for sleeping sickness in humans and nagana in cattle in sub-Saharan Africa. Microscopy and Hi-C data from *T. brucei* have suggested that the 3D nuclear organization is important for RNA maturation[30–32]. In those studies it was found that highly expressed *T. brucei* genes, such as the active VSG gene and tubulin genes, make frequent inter-chromosomal interactions with a locus that produces the noncoding spliced leader RNA that is trans-spliced to the 5´ end of every mRNA and is thus crucial for mRNA maturation[33]. These findings led to a model in which a distinct 3D nuclear organization may enhance the maturation of specific transcripts. Similarly, in mammalian cells, it was observed that nuclear speckles represent major structural hubs that organize inter-chromosomal contacts. Such structures reduce the spatial distance of genes from these speckles, thereby increasing RNA splicing rates[34].

What remains unclear is whether these specific inter-chromosomal interactions identified by microscopy, SPRITE, and 3C-based methods represent special cases or broader principles of global genome organization. It seems plausible that the organization of the genome into distinct inter-chromosomal hubs, e.g., around functionally related DNA elements, may prove especially important in organisms lacking some of the well-studied contact domains such as TADs and loops between promoters and enhancers (E-P loops).

However, most high-resolution 3C-based analyses have been performed in organisms with distinct intra-chromosomal organization, such as mammals[35,36], *Drosophila*[37], and yeasts[38–40], making it difficult to judge the evolutionary importance of inter-chromosomal interactions. In addition, transcription-dependent topologies, such as clustering of genes transcribed by RNAPII, may be transient features difficult to detect in data generated from mixed populations, given that nuclear DNA organization is highly dynamic in mammals[41–44]. Furthermore, the large size of the human, mouse, or *Drosophila* genome make the generation of high-resolution contact maps costly. As a result, most studies do not have the necessary depth to investigate genome-wide DNA contacts at high resolution or require specialized strategies to focus on a subset of genomic regions[45,46].

Here, to investigate the role of inter-chromosomal hubs in the absence of TADs and E-P loops, we generated high-resolution Micro-C contact maps in *T. brucei*. Unlike more complex eukaryotes, *T. brucei* lacks enhancers and CTCF[47], and a previous Hi-C study points to the absence of TAD-like structures[48]. In addition, unusually for a eukaryote, its ~9000 genes transcribed by RNAPII are organized into a few hundred polycistronic transcription units (PTUs). Moreover, transcription initiation of the PTUs is unregulated, removing the interference of dynamic transcription regulation in the interpretation of architectural data[49,50]. These features, in addition to its relatively small genome of 35 Mb, make *T. brucei* an ideal organism to use high-resolution 3C methods to investigate the evolutionary role of inter-chromosomal hub structures and to identify transcription-dependent nuclear topologies. A challenge in analyzing 3D genome organization in less well-studied organisms has been a lack of chromosome-scale genome assemblies, which complicates the analysis of 3C data and often leads to erroneous data interpretation. Particularly, extended gaps, collapsed repeat regions, and incorrectly scaffolded contigs can lead to misinterpretations of 3C data. Despite significant improvements in the quality of the *T. brucei* genome assembly over the last 10 years[48,51,52], the current assembly of the Lister 427 genome (version 10) contains 103 gaps and 74 collapsed repeats. Thus, for this study we generated ultra-long nanopore reads at ~350x coverage, enabling us to greatly reduce the number of gaps and collapsed repeats in the new genome assembly. Next, we established high-resolution Micro-C assays, allowing us to determine the parasite's genome architecture at unprecedented resolution. This revealed an intricate 3D genome organization around polymerase-class specific transcription hubs.

Our observations indicate that not only do RNAPI-transcribed rRNA genes interact with each other, but RNAPII TSSs also come together to form inter-chromosomal transcription hubs. Similarly, we find that RNAPIII-transcribed tRNA genes interact with other tRNA genes, albeit in a more selective manner than RNAPII TSSs.

## Results

### Ultra-long nanopore reads resolve most gaps and collapsed repeats in the *T. brucei* genome assemblies

To investigate the presence of polymerase class-specific inter-chromosomal hub structures, we sought to determine whether regions transcribed by RNAPI, RNAPII, or RNAPIII interact more frequently with themselves than with other regions of the genome. In *T. brucei*, 45 of the 120 annotated RNAPI-transcribed rRNA genes are located in repetitive regions that are collapsed or absent in the genome assembly (Lister 427 version 10). We therefore aimed to fill gaps and correct collapsed regions to their true size using ultra-long Oxford Nanopore Technology (ONT) sequencing reads.

Approximately ~350-fold coverage of the *T. brucei* Lister 427 strain genome was generated to correct the previous fully phased Lister 427 assembly (version 10)[51]. Previous data[51] were merged with six new sequencing runs, with read N50s ranging from ~40 kb to ~440 kb (Supplementary Table 1). The merged data allowed the partial or complete closure of 85 of the 103 gaps, ranging in size from 2 to 50 kb (Fig. 1a, b). To assess the completeness of gap closure, we used our long ONT reads and extracted the first and last 1.5 kb of each read to generate virtual paired reads (VPRs) (Fig. 1c, d and Supplementary Fig. 1a). Since the distance between the paired reads is known from the ONT read length, the distance between genome-aligned VPRs can be used to evaluate whether a gap has been completely closed and whether a collapsed repeat region has been extended to the correct length. Using the VPR analysis, we were able to validate the complete closure of 58 gaps. The remaining gap closures resulted in partial closures or could not be verified by VPRs. Similarly, we used the VPRs to identify 74 regions where assembly size was inconsistent with the size of the ONT reads. Of these 74 regions, we corrected 63 to their correct length, shortening 13 by up to 12 kb and extending the remaining 50 by up to 17 kb (Fig. 1a, b).

In order to accurately map our sequencing data, it is critical to take into account the unusual linear structure of the trypanosome genome. *T. brucei* has a diploid genome with individual chromosomes containing both homozygous core regions and subtelomeric regions with a very high level of heterozygosity. When mapping sequencing data, it is important to represent the highly similar regions only once to avoid mapping sequencing reads to multiple loci. Therefore, we generated a working version of our new genome containing both haplotypes (A and B) of the highly heterozygous subtelomeric regions assembled as separate contigs, but only haplotype A of the diploid core regions.

Importantly for the subsequent analyses, in the new genome assembly, all TSSs, rDNA loci and 95% of the tRNA genes are in correctly assembled regions (i.e., more than 1 kb from an open gap or collapsed region). In addition, 16 of the 21 annotated centromeres are now fully assembled, ranging in size from 9 to 55 kb. These improvements to the *T. brucei* Lister 427 genome assembly significantly enhance its continuity and make it well-suited for genome wide sequencing-based analyses, including 3C-based studies.

## Micro-C reveals organization of RNAPII transcription start sites into inter-chromosomal hubs

Previous Hi-C analysis, in which DNA was crosslinked with formaldehyde and fragmented by a specific restriction enzyme, revealed important features of the *T. brucei* genome organization within the nucleus. For example, while no evidence for TADs was found, inter-chromosomal interactions were detected that appear to enhance trans-splicing of highly expressed genes[30,32,48]. However, the resolution of Hi-C is limited by the cutting frequency of the restriction enzyme used. To detect DNA-DNA interactions at the highest possible resolution and to evaluate the presence of interactions between polymerase-specific transcription start sites, we established Micro-C[35,39] in *T. brucei*. Using an asynchronous culture with 80-90% of cells in interphase, DNA is fragmented with micrococcal nuclease (MNase), which cuts DNA between nucleosomes, allowing a resolution of ~150 bp. Additionally, a second crosslinking step was added to allow a better capture of long-distance interactions and reduce the signal-to-noise ratio[7].

After sequencing, to correctly adjust for the discrepancy between the physical number of genomic regions and the number of contigs in the genome assembly due to the unusual structure of the *T. brucei* genome, we corrected our Micro-C data for ploidy by duplicating interactions from underrepresented subtelomeric regions (for details see Methods and ref. 53).

Visual inspection of the Micro-C data revealed a striking pattern of enriched interactions between RNAPII TSSs, suggesting the presence of distinct chromatin loops and RNAPII hub structures (Fig. 2a). To systematically identify loops, i.e., enriched interactions between distant bins on the same chromosome, we used Mustache[54], a tool for multi-scale detection of chromatin loops from Hi-C or Micro-C data. Using Mustache, we identified 367 loops. Of the 367 loops identified, 269 connected two sites (bins) annotated as TSS. Analysis of the remaining 98 loops revealed that they either connected sites with low read coverage, resulting in high noise, or connected a TSS to a bin that was immediately adjacent to a TSS. Taken together, this analysis suggests that it is predominantly regions with RNAPII TSSs that are involved in loop formation. To confirm this observation, we performed an aggregate analysis, averaging interactions between pairs of RNAPII TSSs located on the same chromosome, and again found that these sites interact much more frequently than control regions (regions of identical size located 20 kb downstream on the same chromosome) (Fig. 2b). Of the 1914 TSS-TSS interactions pairs for which interaction frequencies could be determined, 64.8% had a ≥1.25-fold higher interaction frequency than their corresponding control region.

Classical chromatin loops and most promoter-enhancer interactions occur within chromosomes. However, when we extended our analysis to include pairs of TSSs located on different chromosomes, we still detected enriched interactions between TSSs (Fig. 2c). Of the 14,412 TSS-TSS inter-chromosomal interactions pairs for which interaction frequencies could be determined, 76.8% had a ≥1.25-fold higher interaction frequency than their corresponding control region. Although inter-chromosomal TSS-TSS interactions occurred at a lower frequency than interactions between TSSs located on the same chromosome, these results suggest the formation of inter-chromosomal hub structures where RNAPII transcription start regions cluster (Fig. 2c).

Unlike TSSs in yeast and more complex eukaryotes, TSSs in *T. brucei* are not marked by well-defined nucleosome-depleted regions[55]. Nonetheless, our data were normalized by Iterative Correction[56] (IC) to ensure that the observed TSS-TSS interactions did not result from an artifact related to MNase digestion bias. Additionally, we re-analyzed previously generated Hi-C data, where DNA was digested using the 4-base cutter MboI. While individual TSSs with enriched contacts were only clearly identifiable in our Micro-C data (Fig. 2a), aggregate analyses around all sites of enriched contacts based on Micro-C data (n = 367) also showed enrichment in the Hi-C data (Fig. 2d). The detection of these interactions by Hi-C and Micro-C indicates that the observed interactions between TSSs are not due to biases in MNase digestion.

## RNAPII TSSs segregate into distinct compartment domains

The best-studied mechanism for chromatin loop formation in metazoans involves the cooperation of CTCF, a DNA-binding protein, and cohesin, a ring-shaped protein complex, in a process called loop extrusion[10]. In this process, cohesin complexes are thought to actively extrude loops of chromatin through their ring-like structure until they encounter boundary elements such as CTCF, where extrusion is blocked. However, non-metazoans including *T. brucei* lack CTCF[47]. Furthermore, in contrast to studies in other non-metazoans such as budding and fission yeasts[38–40] or in closely related species such as *T. cruzi*[57,58], studies on the nuclear architecture in *T. brucei* have not identified any TAD-like structures[48].

However, to better understand the nature of the observed inter-chromosomal RNAPII TSS hub structures, we used the new Micro-C data and performed a systematic analysis to identify compartment domains and other contact domains based on changes in eigenvector value and insulation score, respectively.

Eigenvectors calculated from the correlation matrix at 10 kb resolution using FAN-C[8] suggested non-exclusive compartmentalization at three levels. As described previously[48], we found that the transcribed chromosome core regions and the untranscribed subtelomeric

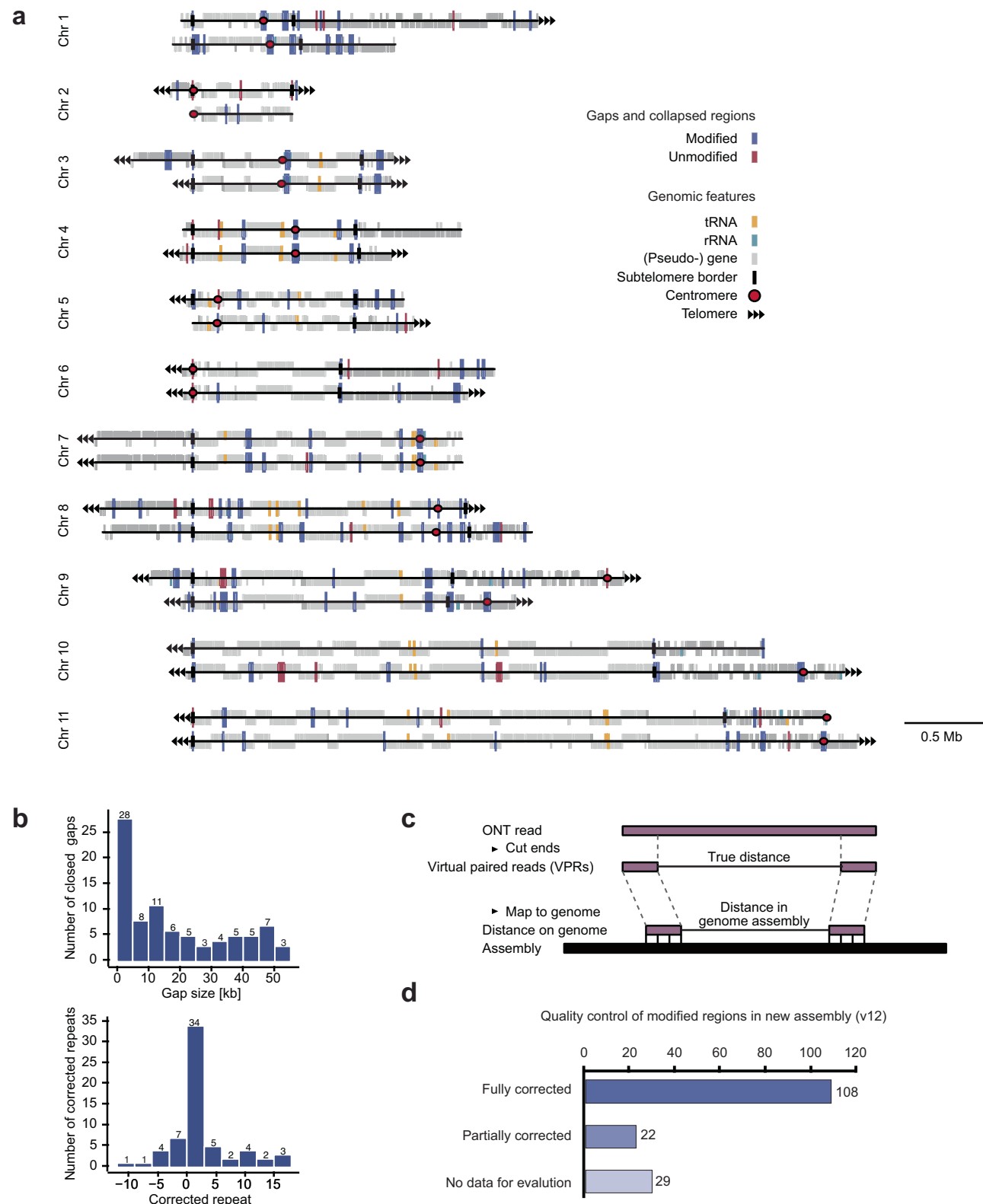

**Fig. 1 | A highly contiguous *T. brucei* genome assembly was generated with ultra-long nanopore reads.** **a** An overview of the features and regions on the megabase chromosomes of the Lister 427 *T. brucei* assembly that have been modified. Allele 'A' and 'B' are shown on the top and bottom of each chromosome, respectively. Modified regions (gaps or collapsed regions, fully or partially corrected) are marked in blue. Regions with open gaps or collapsed repeats that could not be corrected and remained unmodified are marked in red. **b** Sizes of closed gaps (top) and change in size of corrected repeat regions (bottom). **c** Illustration of our strategy to confirm the correctness of the modifications made to the genome. Modified regions are considered corrected if there are at least 2 VPRs with less than 5 kb difference between the distance where the VPRs map on the assembly and their distance based on ONT read. **d** Correctness evaluation of modified regions in the new genome assembly (version 12).

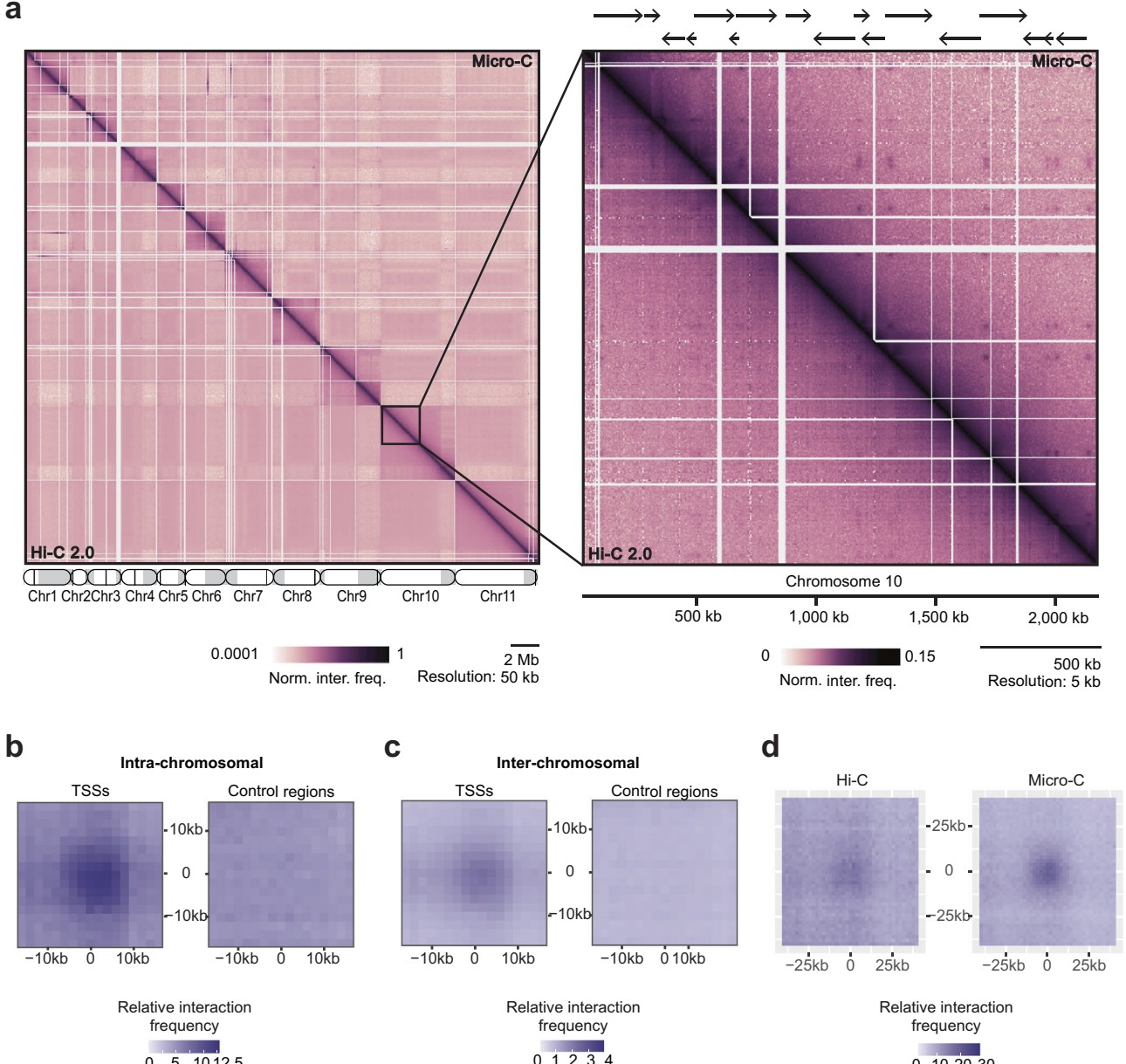

**Fig. 2 | Micro-C reveals interactions among RNAPII transcription start sites.**
**a** IC-normalized and ploidy-corrected heat maps of Hi-C (bottom left) and Micro-C (top right) of megabase chromosomes 1-11 (haplotype A) at 50 kb resolution (left) with each chromosome displayed with the core region in white, subtelomeric regions in light gray and centromeres displayed as black lines; part of chromosome 10 of the same heat maps is displayed at 5 kb resolution (right), with polycistronic transcription units of chromosome 10 illustrated as black arrows (top).
**b** Aggregate analysis of Micro-C data around pairs of intra-chromosomal TSSs

($n = 2342$ TSS-TSS pairs); control loci are obtained from the same pairs of loci shifted downstream by 20 kb. **c** Aggregate analysis of Micro-C data around pairs of inter-chromosomal TSSs ($n = 20,008$ TSS-TSS pairs); control loci are obtained from the same pairs of loci shifted downstream by 20 kb. **d** Aggregate peak analysis of Hi-C and Micro-C data around chromatin loops (i.e., pairs of bins with high inter-action frequency) identified in the Micro-C matrix using Mustache (with a *p*-value threshold of 0.02 and a sparsity threshold of 0.85; $n = 367$).

regions fold into distinct compartments. In addition, the eigenvector values suggest that in some cases the two chromosome arms separated by the centromere may fold into distinct compartments. Third, our data suggest that RNAPII TSSs form a compartment distinct from the rest of the core regions (Fig. 3a and Supplementary Fig. 6).

Next, we used FAN-C[8] to calculate the insulation score, a measure of the average frequency of interactions that extend beyond a given bin (here 10 kb). Typically, a low insulation score marks the boundaries of contact domains, including TADs. However, low insulation scores can also be caused by analysis artifacts, such as regions of low mapp-ability caused by repeat regions, gaps in the genome assembly, or

incorrectly scaffolded contigs. Our data suggest that low insulation scores correlate with low read mappability in all but one region (Fig. 3b and Supplementary Fig. 7). Interestingly, this region on chromosome 8 is flanked by two sites that represent boundaries between PTUs, con-tain tRNA genes, and are enriched in cohesin (Fig. 3b), similarly to what has been found in closely-related species[58].

Thus, our Micro-C data suggest that while the *T. brucei* genome appears to be organized into distinct compartmental domains, with almost all regions of low insulation coinciding with sites of low mappability, our data did not provide evidence for the presence of contact domains separated by low insulation scores, such as TADs and TAD-like structures, except in one case.

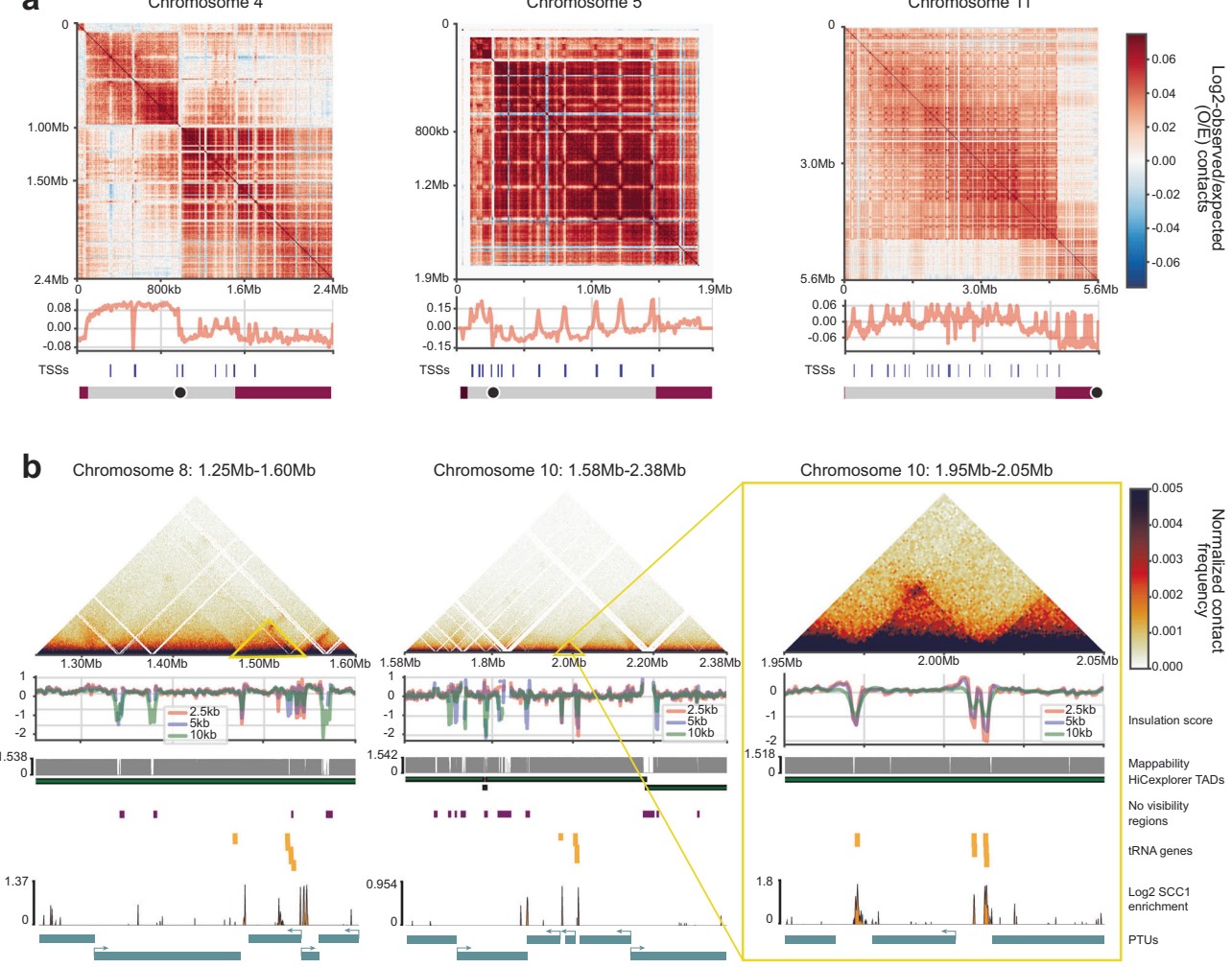

**Fig. 3 | Transcription start sites localize to distinct kilobase-scale compartments. a** Examples of correlation matrices for chromosomes 4, 5 and 11 (haplotype A) and corresponding second eigenvector calculated from the Micro-C data analyzed with the distiller pipeline at 10 kb resolution (using FAN-C[8]). Each chromosome is displayed under the eigenvector plot with the core region in grey and subtelomeric regions in dark red; centromeres are represented as black dots.

**b** Examples of domain boundaries identified manually with insulation scores calculated from the Micro-C data analyzed with the distiller pipeline at 2.5, 5 and 10 kb resolution using FAN-C[8] (top); mappability of the region, TAD-like domains identified with HiCexplorer[106], no visibility regions (i.e., regions where signal was removed by IC normalization, for details see Methods), tRNA genes, SCC1 ChIP-seq fold enrichment and PTUs are displayed (bottom).

## Interactions among RNAPII TSSs are independent of cohesin-mediated loop extrusion

The clustering of RNAPII TSSs into distinct compartments, the lack of evidence for the existence of TAD-like structures in *T. brucei*, and the relatively high frequency of inter-chromosomal interactions between TSSs argue against cohesin-mediated loop extrusion as the main driver in establishing the observed TSS hub structures.

However, since cohesin is often found to be enriched at TSSs and cohesin-mediated chromatin loops are frequently anchored at TSSs[59], we asked whether cohesin contributes to the formation of the observed TSS hubs. Depletion of cohesin followed by Micro-C assays would be the most direct way to determine if it contributes to hub formation, but depletion of the major cohesin subunit SCC1 by RNAi leads to rapid cell death in *T. brucei*[60,61]. Instead, the distribution of cohesin across the *T. brucei* genome was reassessed using existing SCC1 ChIP-seq data[48] and our improved genome assembly. Our re-analysis revealed that SCC1 is enriched at transcription termination sites (TTSs) but depleted at TSSs (Fig. 4). Because PTUs are found on either strand of the genome, their TTSs can be convergent (cTTS) or, when PTUs are oriented head-to-tail, they can be located upstream of

adjacent start sites as single TTS (sTTS). We found cohesin to be enriched at both convergent and single TTSs. In fact, cohesin levels gradually increase along PTUs, with the highest cohesin levels at the TTSs (Fig. 5a), as if its distribution is dependent on elongating RNAPII. In contrast, it is the TSS at the beginning of the PTU that engage in interactions with other TSSs (Fig. 5a). These observations suggest that a role of cohesin in the establishment of the observed TSS hubs is unlikely.

The fact that such interactions were not observed for RNAPII TTSs or for the body of PTUs (Figs. 4 and 5b) suggests that the specialized chromatin structure associated with RNAPII TSSs may be important for the formation of these RNAPII TSS hubs. Unlike in most eukaryotes, in *T. brucei* RNAPII TSSs are not marked by well-positioned nucleosomes and narrow peaks of H2A.Z containing +1 nucleosomes[55]. In contrast, *T. brucei* RNAPII TSSs exhibit high levels of the histone variants H2A.Z and H2B.V, histone acetylation at positions H4K2, H4K5, and H4K10, and tri-methylation at position H3K4. These histone variants and modifications are enriched across regions of ~10 kb in width, encompassing the first few genes of every PTU[62–66]. We re-analyzed previously generated H2A.Z MNase-ChIP-seq data[55] using our improved genome

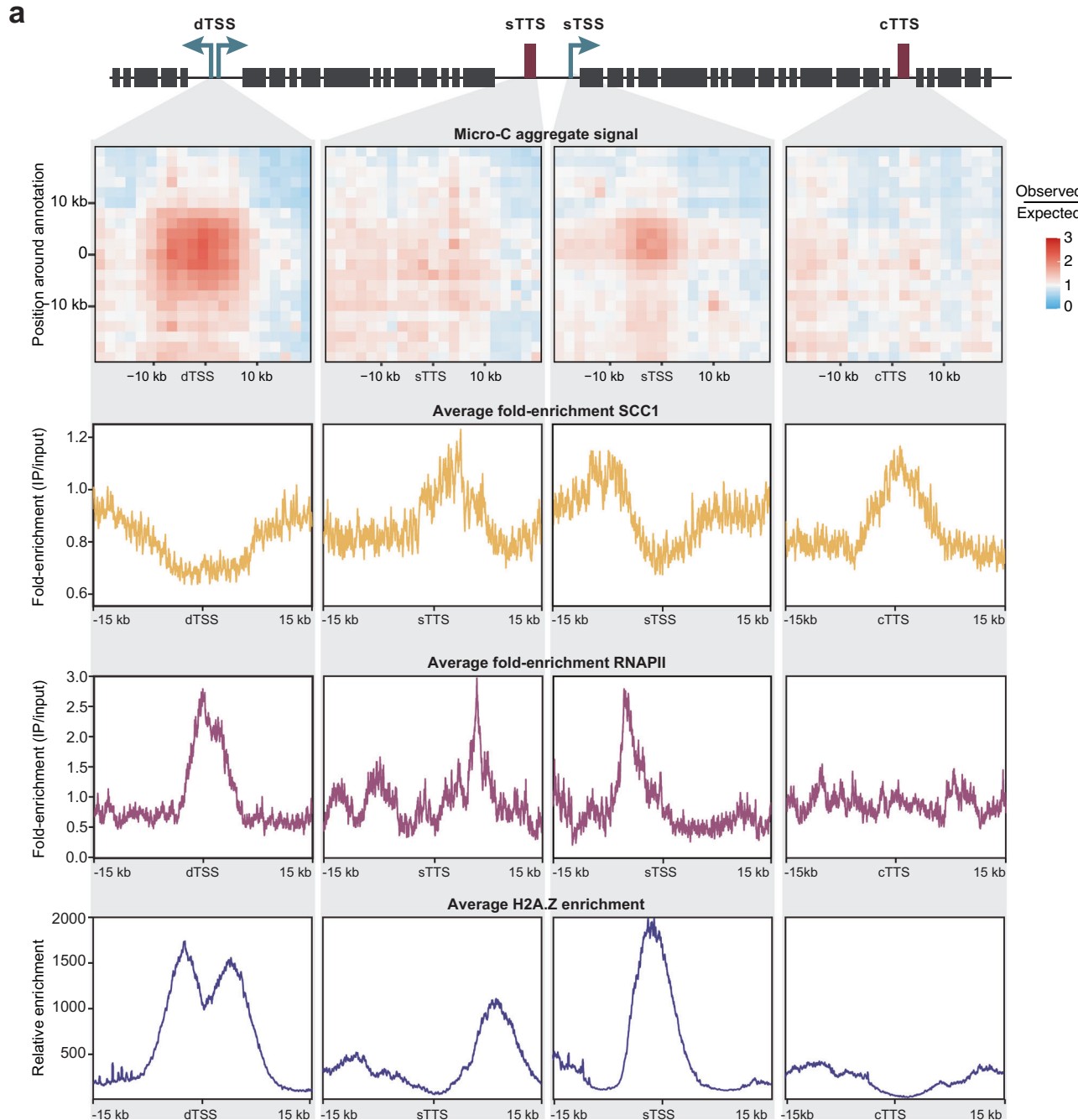

**Fig. 4 | Cohesin is depleted from sites engaging in TSS-TSS interactions. a** Top panel: schematics of polycistronic transcription unit types (head-to-head and head-to-tail) with divergent and single transcription start sites (dTSS and sTSS, in green) and single and convergent transcription termination sites (sTTS and cTTS, in red). Bottom panels: for each of the sites, aggregate Micro-C signal (observed/expected), metaplot of SCC1 ChIP-seq[48] fold enrichment, RNAPII ChIP-seq[66] fold enrichment and H2A.Z MNase-ChIP-seq[55] enrichment.

assembly. This analysis revealed that a much higher percentage of Micro-C contact anchor sites overlapped with peaks of H2A.Z MNase-ChIP-seq peaks (69.8%) than with SCC1 ChIP-seq peaks (1.4%; Fig. 5c). This finding suggests that it may be the unique chromatin structure found at TSSs that drives clustering of TSSs rather than cohesin-mediated loop extrusion.

### RNAPI transcribed genes cluster together selectively

RNAPII transcription in *T. brucei* is unregulated and constitutively active[49,50]. In contrast, RNAPI-transcribed genes coding for the major surface antigens, the variant surface glycoproteins (VSGs), are highly regulated. More than 2000 VSG-encoding genes are present in the *T. brucei* genome, but only one of these VSG genes is transcribed at any given time[32]. Moreover, FISH and Hi-C analyses indicate that the majority of transcriptionally repressed VSG genes cluster into several distinct loci[48,67]. In contrast, the single actively transcribed VSG gene is located away from these clusters of transcriptionally inactive VSG genes[30].

Perhaps relevant to the silencing of RNAPI-transcribed VSG genes is the fact that not all rRNA genes are actively transcribed simultaneously in other organisms and that their activation can be selective. Some rRNA genes are more transcriptionally active than others in

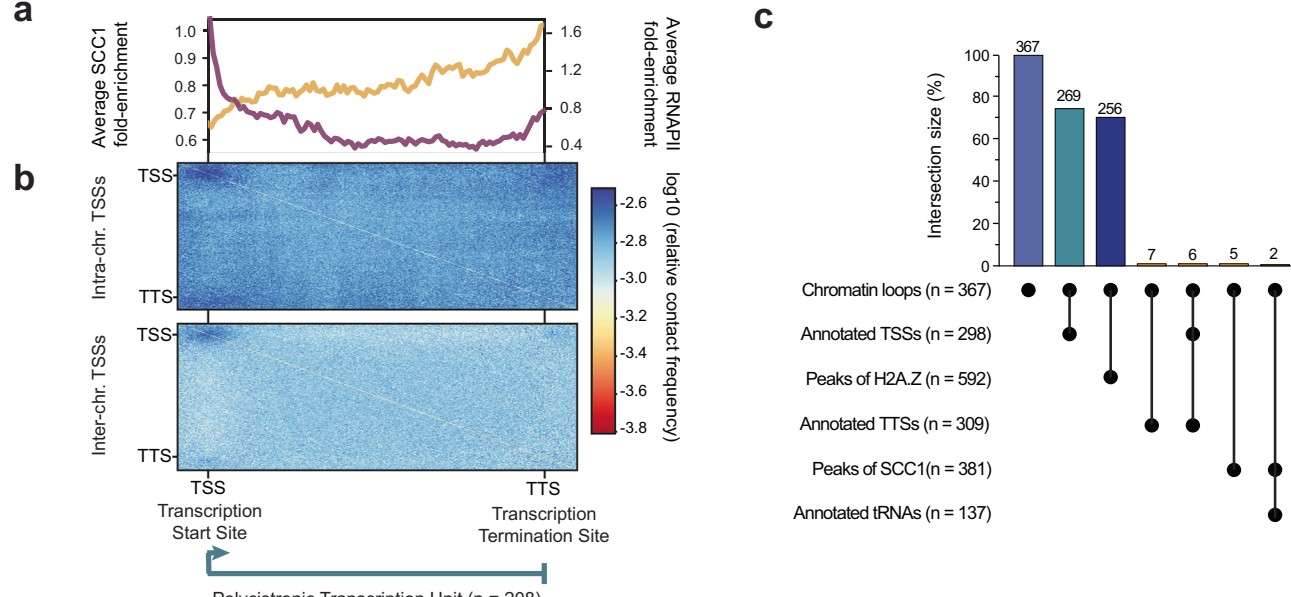

**Fig. 5 | Cohesin levels gradually increase along PTUs. a** Metaplot of SCC1 ChIP-seq[48] fold-enrichment (yellow) and of RNAPII ChIP- seq[66] fold enrichment (purple) along polycistronic transcription units (n = 208). **b** Aggregated plot of interaction frequencies calculated from Micro-C data between pairs of PTUs (n = 208). PTUs were scaled to the same length and extended by 10% up- and downstream. **c** UpSet plot showing the chromatin loops (i.e., pairs of bins with high interaction frequency) identified in the Micro-C matrix at 5 kb resolution (n = 367), where both loci of a pair overlap with annotated TSSs (n = 298), H2A.Z MNase-ChIP-seq[55] peaks (n = 592; see Methods), annotated TTSs (n = 309), SCC1 ChIP- seq[48] peaks (n = 381; see Methods), and/or annotated tRNA genes (n = 137).

mammalian cells, and these active rRNA genes tend to cluster together within the nucleolus[68–71]. Similarly, in *T. brucei*, promoters of different rRNA genes show differential activity[72,73].

To determine whether interactions between *T. brucei* rRNA genes occur across the genome, we plotted our Micro-C data in a 4C-like manner using RNAPI-transcribed rDNA regions as the selected viewpoints. Since rRNA genes are often arranged in tandem arrays along chromosomes, we defined rDNA regions as groups of adjacent rDNA loci separated by less than 10 kb. This resulted in the assignment of 16 rDNA regions distributed across the 22 megabase-sized chromosomes. Significance was assigned to virtual 4C peaks using a two-sample Kolmogorov–Smirnov test (KS-test; see Methods). However, since virtual 4C peak sizes are influenced by factors such as the size of the viewpoint and the mappability of viewpoint and target regions, we also examined interactions qualitatively.

Our analysis revealed a network of interactions in which rDNA loci interact preferentially in specific groups. Specifically, the rDNA loci on chromosomes 1, 3, and 7 form one cluster, while two rDNA genes on the subtelomeric region of chromosome 11 A interact with the rRNA clusters on chromosomes 9 and 10 (Fig. 6a). Similar to the polymerase class-specific clustering we saw for RNAPII TSSs, we observed that the RNAPIII-transcribed 5S rRNA genes located on chromosome 8 interact only minimally with RNAPI-transcribed rRNA loci, and we suggest that these interactions are a consequence of the spatial clustering of rRNA genes in the nucleolus. The repetitive nature of rRNA gene arrays prevented us from conclusively assessing patterns of selective transcription. Thus, it remains to be seen whether the selective clustering of rDNA into different hubs correlates with the transcriptional activity of these loci.

### Micro-C reveals pairwise interactions between RNAPIII-transcribed tRNA genes

In most eukaryotes, including *T. brucei*, tRNA genes are scattered throughout the genome and are often found in arrays on multiple chromosomes[74–76]. Despite this dispersed linear organization, previous chromosome conformation and nuclear localization data[22–24] suggest that these genes associate in 3D space within nuclei. Such clustering of tRNA genes has been hypothesized to increase the efficiency of their transcription by concentrating the necessary transcription factors and RNAPIII machinery in specific nuclear regions[77]. *T. brucei* RNAPIII transcribes tRNA genes, snRNAs located in close proximity to tRNA genes, and 5S rRNA genes[78–81]. To identify any prominent spatial contacts between RNAPIII-transcribed genes, we performed a virtual 4C analysis using every cluster of tRNA genes as separate viewpoints. tRNA clusters were defined as groups of adjacent tRNA genes where each gene is separated from the next by less than 10 kb. In contrast to our analysis of RNAPII- and RNAPI-transcribed genes, we found that while approximately 25% of the tRNA regions contacted one or more RNAPIII-transcribed regions, the others did not appear to engage in DNA-DNA contacts with other RNAPIII-transcribed loci (Fig. 6b, bottom track, and Supplementary Fig. 8).

### Micro-C suggests that centromeres form multiple clusters

Given the unusual organization of the genome around polymerase class-specific hubs, we decided to extend our analysis to study the organization of other important genomic features often found to interact in specific clusters, such as centromeres.

To characterize centromere-centromere interactions in more detail, we improved the assembly of centromeric sequences and annotated the centromeres using information from the kinetochore protein KKT2 ChIP-seq data[82] or the presence of centromere-specific CIR147 repeats[83,84]. Using these data, we were able to annotate the centromeres of 21 of the 22 megabase-sized chromosome homologs. No KKT2 signal was detected on chromosome 10 A. Our annotations indicate that centromere regions vary in length from 3 to 50 kb and are composed of different classes of repeat elements, as suggested based on topoisomerase II cleavage[85] and confirmed by our consensus repeat sequence analysis (Table 1 and Supplementary Table 3). Because the repetitive nature of centromeric DNA makes it difficult to reliably map reads to specific centromeric loci, we generated virtual 4C plots for

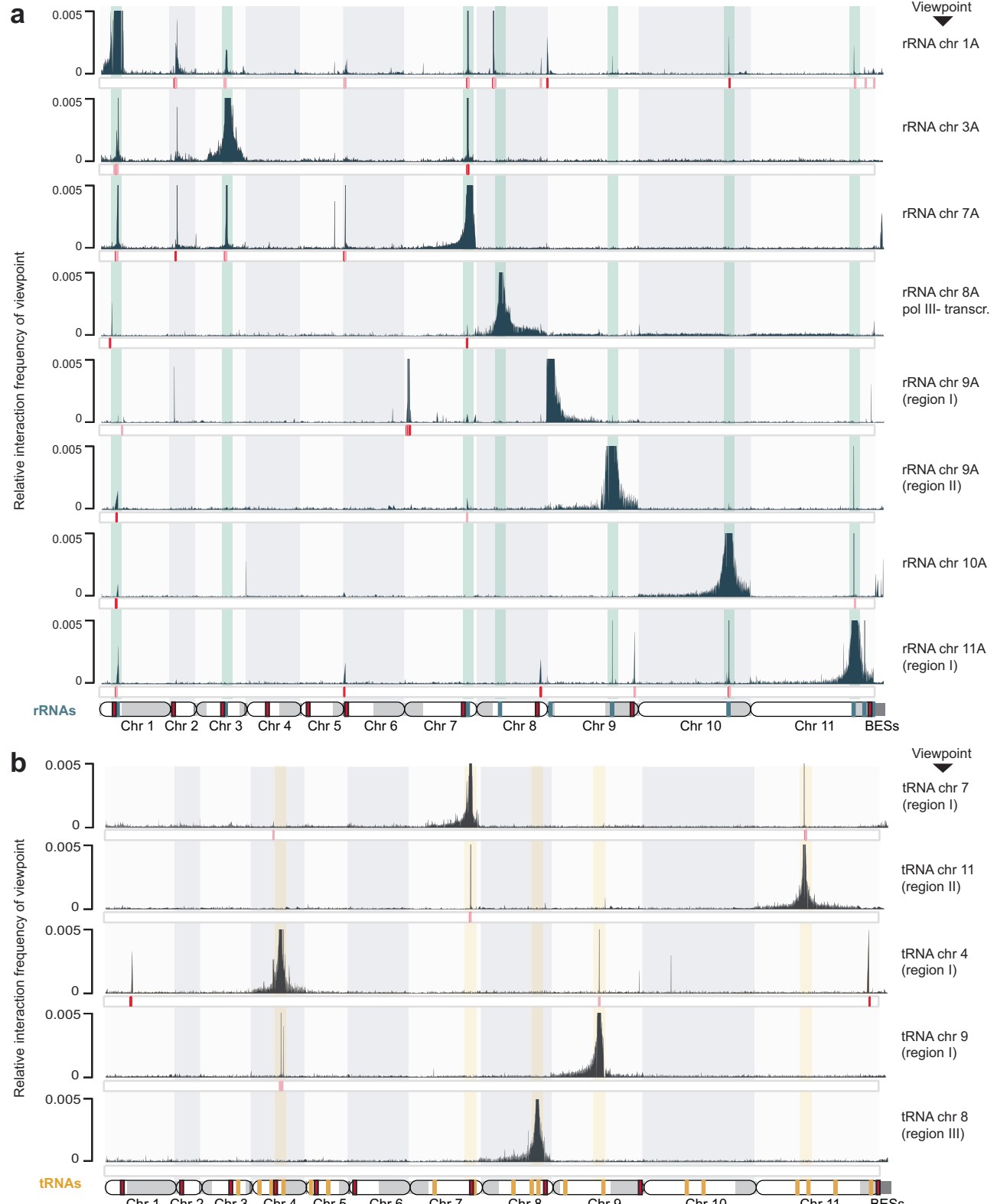

**Fig. 6 | rRNA and tRNA gene loci form polymerase class-specific clusters. a** Top: Virtual 4C interaction profiles based on Micro-C data (10 kb resolution) showing the interaction frequencies of rDNA regions (defined as groups of adjacent rDNA loci separated by less than 10 kb) used as viewpoints with the Tb427 genome version 12, haplotype A. Significant interactions as calculated using a two-sample KS-test were marked below each track (pink: $p < 0.05$; red: $p < 0.01$). Bottom: rDNA regions are highlighted in green; red boxes represent centromeres; each chromosome is displayed with the core region in white and subtelomeric regions in light gray. **b** Top: Virtual 4C interaction profiles based on Micro-C data (5 kb resolution) showing the interaction frequencies of tRNA regions (defined as groups of adjacent tRNA loci separated by less than 10 kb) used as viewpoint with the Tb427 genome version 12, haplotype A. Significant interactions as calculated using a two-sample KS-test were marked below each track (pink: $p < 0.05$; red: $p < 0.01$). Bottom: tRNA regions are highlighted in yellow; red boxes represent centromeres; each chromosome is displayed with the core region in white and subtelomeric regions in light gray.

**Table 1 | Size, location, period size and class of centromeres in the newly assembled genome Lister 427 version 12**

| Chromosome | Size (bp) | Contig type | Period size (bp) and class |
|---|---|---|---|
| 1A | 43,790 | core | 146, class I |
| 1B | 50,856 | core | 146, class I |
| 2A | 14,757 | core | 29, class II |
| 2B | 8800 | core | 29, class II |
| 3A | 24,693 | core | 120, class IV |
| 3B | 24,343 | core | 120, class IV |
| 4A | 33,158 | core | 148, class III |
| 4B | 54,643 | core | 148, class III |
| 5A | >21,486* | core | 147, class III |
| 5B | 20,864 | core | 147, class III |
| 6A | >3745* | core | 58/136, class I |
| 6B | >3029* | core | 58/136, class I |
| 7A | 39,869 | core | 30, class II |
| 7B | 39,480 | core | 30, class II |
| 8A | 19,090 | core | 147, class III |
| 8B | 18,625 | core | 147, class III |
| 9A | >2741* | 3′ subtelo | 39, class I |
| 9B | 50,482 | 3′ subtelo | 39/78, class I |
| 10B | 18,287 | 3′ subtelo | 49, class I |
| 11A | >3806* | 3′ subtelo | 49, class I |
| 11B | 43,917 | 3′ subtelo | 20/59, class I |

Centromeres that are not fully assembled are marked with*.

each centromere (Fig. 7a) and repeated the analysis using 5 kb regions immediately upstream or downstream of the bins containing the centromere repeats as viewpoints (Supplementary Fig. 9 and 10). We reasoned that interactions of flanking regions with a specific centromere in a distance-dependent manner would strongly suggest that these centromeres interact. Using this approach, we found that centromeres 1, 2, 3, and 7 clustered together (Fig. 7a) and that centromere 4 clustered with centromere 5 (Fig. 7a and Supplementary Figs. 9, 10). Moreover, we used fluorescence in situ hybridization (FISH) with probes hybridizing to the CIR147 repeats at centromeres 4, 5 and 8 to quantify the number of clusters in the nucleus of *T. brucei*. In the absence of centromere clustering, we would expect to detect as many foci by FISH as there are physical copies in the genome. In the case of CIR147 repeats, we would therefore expect 6 foci in G1 cells and 12 in dividing cells. Instead, we detected <4 foci in the majority of cells (91%; Fig. 7b). These results suggest that centromeres 4, 5 and 8 are indeed colocalizing. As control we used FISH to visualize the region transcribed into spliced leader RNA, a single repeat region located on both homologs of chromosome 9. Here, we find the number of FISH loci to match the number of genomic sites, indicating that these two sites do not cluster (Supplementary Fig. 11). Taken together, these findings suggest that not only transcribed regions but also centromeres cluster into specific hubs (Fig. 7c).

## Discussion

Inter-chromosomal interactions play an important role in chromatin domain formation, gene regulation and genome stability[86]. However, previous studies have predominantly focused on intra-chromosomal structures and their impact on transcriptional regulation. Given the unique organization of trypanosome genes into a few hundred transcription units and their constitutive transcriptional activity[49,50], this unicellular parasite represents an excellent model to explore the principles of inter-chromosomal interactions.

To study the nuclear architecture of *T. brucei* in detail, we generated a highly contiguous genome assembly, closing most gaps and

correcting 85% of the collapsed repeats in the 11 megabase-sized chromosomes. We then performed a high-resolution Micro-C analysis of bloodstream *T. brucei*, generating >170 million uniquely mapping reads. Given the *T. brucei* genome size of 35 Mb and the human genome size of 3.3 Gb, the resolution of our study is comparable to a study with ~1.6 trillion reads of the human genome when analysing inter-chromosomal interactions.

On a global scale, based on eigenvector values, our data suggest a compartmentalization of the *T. brucei* genome at three levels: 1) The transcribed chromosome core regions and the untranscribed sub-telomeric regions fold into distinct compartments, as we had previously observed[48]. 2) In some cases, the centromeres appear to act as compartment boundaries, with chromosome arms folding into distinct compartments. 3) RNAPII TSSs form a compartment that is distinct from the rest of the core regions. Given that the entire core regions of chromosomes are transcribed in *T. brucei*, we did not expect the core regions to segregate into A and B compartments as described in other organisms. However, we were surprised to find that RNAPII TSSs appear to segregate into a separate compartment, and we propose that this may be a result of the RNAPII TSSs clustering.

To detect contact domains other than compartments, we used our Micro-C data to calculate insulation scores. As in a previous study[57], we detected numerous sites with low insulation scores, marking putative contact domain boundaries. However, we also found that most of these low insulation score sites coincided with sites of low read mappability, implying that the interaction frequencies calculated for these sites are not reliable. Thus, while the output of the contact domain callers should be treated with caution, as it may simply be an artifact caused by low mappability regions, some of the many low mappability sites may contain repetitive regions that represent bona fide contact domain boundaries. One way to address this issue in the future would be to perform Pore-C analyses[87]. The longer reads generated in such studies should address the issue of low unique mappability. In summary, while we think it is possible that future studies provide evidence for the existence of contact domain boundaries, we see no evidence that these domains are generated by loop extrusion, a feature of TAD-like structures.

The high resolution of our dataset revealed both conserved and unusual polymerase class-specific inter-chromosomal hubs. While interactions among RNAPI-transcribed genes within the nucleolus appear to be ubiquitously conserved, and interactions among RNAPIII-transcribed tRNA genes have been described in many organisms[22–24,58], we also observed an extensive pattern of inter-chromosomal RNAPII TSS interactions. Our observations are consistent with previous immunofluorescence analyses of RNAPII[88] and H4K10ac, a histone mark enriched at RNAPII TSSs[62], which revealed a punctate nuclear pattern suggesting clustering. In other organisms, interactions among RNAPII-transcribed genes have been observed mainly for specific gene subsets associated with increased splicing in nuclear speckles[20,30,89].

Taken together, our Micro-C data suggest that intra-chromosomal interactions leading to organization into compartments enclosing the transcribed core regions or the transcriptionally repressed sub-telomeric regions provide the global genome organization. In contrast, inter-chromosomal interactions provide a much finer additional level of interactions between specific loci, such as RNAPII TSSs, rRNA genes, or between the single actively transcribed VSG gene and the SL array, that may be important for RNA transcription and processing.

Although our data revealed the presence of intra- and inter-chromosomal clustering of RNAPII TSSs, because we measured pairwise rather than multivalent interactions, the true size of these hubs and their variability between cells remains unknown. For example, although our data suggest that most TSSs interact with multiple other TSSs, it is still unclear whether these interactions occur simultaneously or involve different loci in different cells. Furthermore, the driving factors behind these inter-chromosomal hubs remain unknown. One

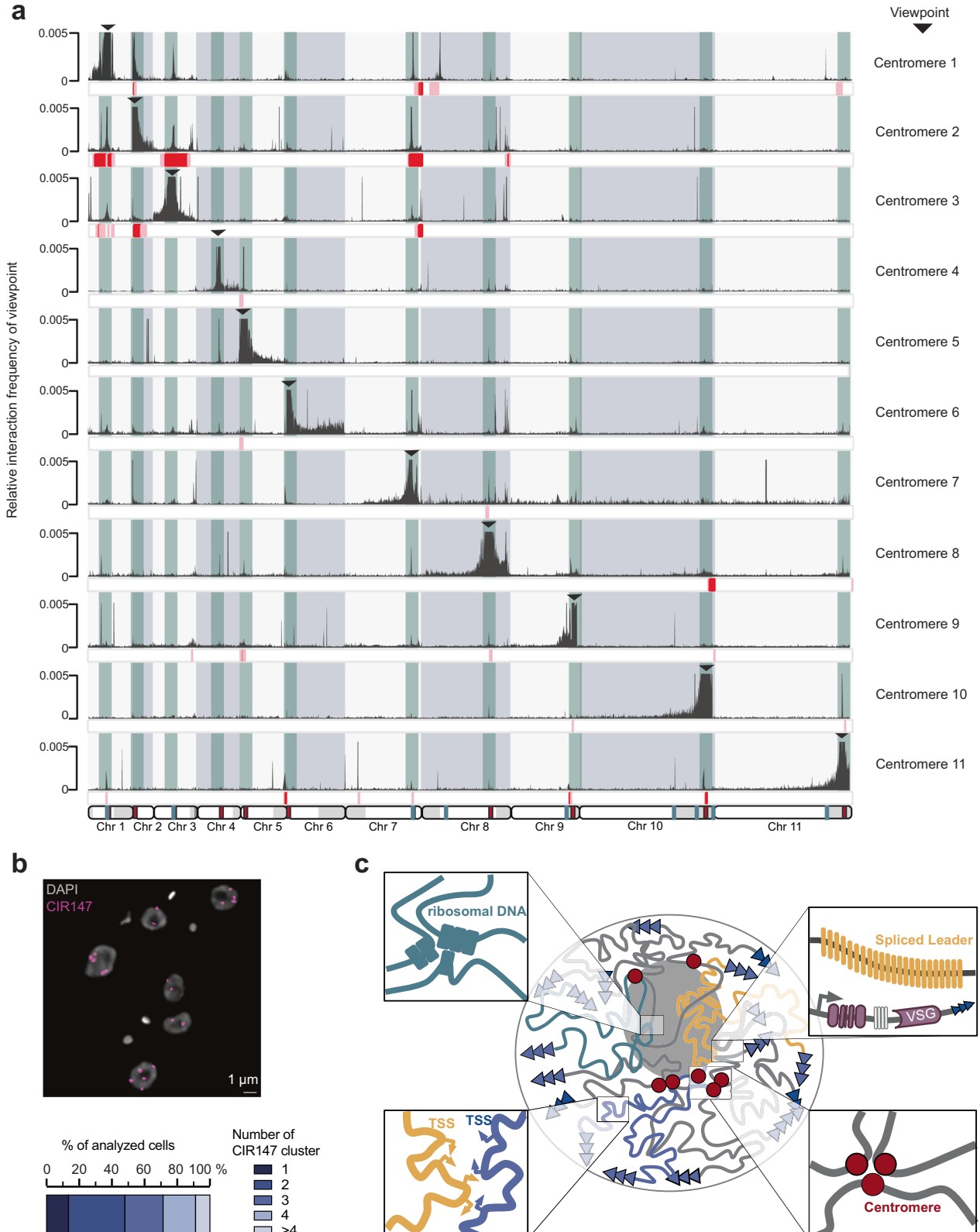

**Fig. 7 | *T. brucei* genome form widespread inter-chromosomal interactions with centromere clustering. a** Top: Virtual 4C interaction profiles based on Micro-C data (10 kb resolution) showing the interaction frequencies of a subset of centromeres used as viewpoints (black arrowheads and green background) with the Tb427 genome version 12, haplotype B. Significant interactions as calculated using a two-sample KS-test were marked below each track (pink: $p < 0.05$; red: $p < 0.01$). Bottom: Each chromosome is displayed with the core region in white and subtelomeric regions in light gray. Centromeres are marked by red rectangles; rDNA loci are marked in green. **b** Fluorescence in situ hybridization (FISH)-based quantification of CIR147 repeats (located on chromosomes 4, 5 and 8) clusters ($N = 1$; analyzed nuclei: 171). Cells displaying no signal were excluded. **c** Model of *T. brucei* genome organization. *T. brucei* chromosomes make frequent centromere-centromere interactions, interactions between RNAPII TSSs, the active VSG gene and the splice leader locus, rRNA genes and some tRNA genes (not displayed).

hypothesis is that the accumulation of RNA polymerases and/or nascent RNAs at TSSs may facilitate the formation of transcriptional condensates, as observed in other species[25–29]. Alternatively, the distinct profile of histone variants and modifications at TSSs may influence compartmentalization[62,65,66,90]. We have previously shown that reduced H2A.Z deposition at TSSs does not affect *T. brucei* RNAPII PTU transcript levels, but instead shifts transcription initiation to more accessible sites in chromatin[65]. Whether such a shift affects the level of interaction between TSSs remains to be seen.

Despite the extensive network of inter-chromosomal interactions, the high frequency of intra-chromosomal interactions observed in this study suggests some degree of chromosomal territoriality. Because our data were generated from asynchronous cells, our results are consistent with a scenario in which chromosomes occupy distinct territories during mitosis and subsequently undergo global rearrangements[91]. Another possibility is that chromosomes move within the nucleus, frequently interacting with intra-chromosomal regions while dynamically establishing contacts with other chromosomes.

In this study, we focused on bloodstream form parasites, the stage that infects the mammalian host. Although Hi-C data exist for insect-stage parasites[30], no systematic comparison of genome architecture has been performed for this stage. Thus, it remains to be seen how the genome organization changes during the life cycle of the parasite. Furthermore, in this and previous studies, we have focused on the diploid set of the 11 megabase-sized chromosomes. However, *T. brucei* also contains 5-6 intermediate chromosomes and ~100 minichromosomes. Because the intermediate and minichromosomes are not yet fully assembled, they have been excluded from genome architecture analyses. Whether and how this affects genome architecture analyses remains to be determined in future studies.

Our results highlight the importance of studying genome organization in evolutionarily divergent organisms, particularly those with small genomes where deep sequencing is affordable, and reveal an intricate pattern of inter-chromosomal DNA-DNA interactions.

# Methods

## *Trypanosoma brucei* cell culture and growth

In this study, two cell lines were used: *T. brucei* wildtype (WT) Lister 427 bloodstream form; and a double-selection *T. brucei* cell lines derived from the Lister 427 bloodstream-form MITat 1.2 isolate[92]. Both cell lines were grown in HMI-11 medium at 37 °C and 5% $CO_2$. In the latter cell line, a neomycin resistance gene in bloodstream-form expression site (BES) 17 and a puromycin resistance gene in BES 1 allowed the selection for a homogenous cell population that expressed either *VSG-2* from BES 1 or *VSG-13* from BES 17. This cell line was grown in HMI-11 supplemented with either 20 µg/mL neomycin (also referred to as N50 cells) or 1 µg/mL puromycin (referred to as P10 cells).

## gDNA extraction for ONT sequencing

For nanopore sequencing, six sequencing libraries were obtained from P10 cells or *T. brucei* wildtype Lister 427 cells. For each library, gDNA extraction was performed using High Molecular Weight DNA Extraction Kit from Cells and Blood (NEB #T3050). The samples were sequenced using a GridION X5 or PromethION 24 platform on a FLO-PRO114M flowcell using MINKNOW software. Sequencing reports and details of each sequencing libraries are shown in Supplementary Table 1.

## Closing gaps and expanding repeats in the assembly

First, an assembly error in BES 2 of the previously fully phased version 10 of the *T. brucei* Lister 427 strain genome assembly[51] was corrected and the contigs names were shortened to generate the version 11 of the assembly (Tb427v11). In order to close gaps and expand collapsed repeats in the Tb427v11 assembly, SAMBA[93] was run with the ONT

reads generated in this study (Supplementary Table 1) and the ONT reads from Girasol et al.[94]. It was confirmed that small contigs are not oriented the wrong way around, an error commonly found in assemblies scaffolded using Hi-C data (Supplementary Fig. 1a–c)[93]. Further, duplicated sequences next to gaps were removed (Supplementary Fig. 2a–d), collapsed repeats were converted to gaps that were successively replaced with the expanded sequences using SAMBA (Supplementary Fig. 3), and gaps in homozygous regions were closed by running SAMBA separately on alleles A and B. 'samtools stats' was used to compute the error rates for each newly generated sequence and all sequences from the Tb427v11 assembly (Supplementary Fig. 4). The step-by-step approach is detailed in Supplementary Note 1.

The newly assembled regions were annotated using Companion[95]. To transfer the annotations from unmodified regions of the Tb427v11 assembly to the improved one, the original annotations' coordinates were transformed based on the cumulative size difference of the modified regions. To avoid read-alignment ambiguities arising from both alleles of a homozygous region being present in an assembly, a working version of the assembly was generated using only one allele (allele A) of each homozygous core region and both alleles of all heterozygous subtelomeric regions, defined as the version 12 of the assembly (Tb427v12).

## Centromere annotation and consensus repeat sequence analysis

KKT2 ChIP-seq data[82] were mapped to the Tb427 version 12 (Tb427v12) genome build using Bowtie2[96] (version 2.4.2), with duplicate reads removed using SAMtools[97]. KKT2 enrichment was identified using MACS2[98] (version 2.2.7.1) broad peak call. Each main chromosome exhibited a large KKT2 enrichment cluster, ranging from approximately 3 kb to 50 kb. The enrichment patterns observed between both biological replicates were used to determine the centromere locations. Consensus sequences on each *T. brucei* chromosome were determined using Tandem Repeats Finder[99].

## Micro-C

The Micro-C protocol was adapted from Krietenstein et al.[35]. We generated 3 biological replicates and 2-3 technical replicates for each biological replicate for a total of 8 replicates. Cells in log phase (0.6-$1 \times 10^6$ cells/mL) were collected and resuspended in 12.5 mL of 1X trypanosome dilution buffer (TDB; 0.005 M KCl, 0.08 M NaCl, 0.001 M MgSO4, 0.02 M Na2HPO4, 0.002 M NaH2PO4, 0.02 M glucose) per 100 million cells. Cells were fixed with 1% formaldehyde for 20 min at room temperature by adding 1.25 mL of freshly made formaldehyde solution (50 mM HEPES-KOH (pH 7.5), 100 mM NaCl, 1 mM EDTA (pH 8.0), 0.5 mM EGTA, 11% formaldehyde). The crosslinking reaction was quenched by adding 2.5 M Tris buffer (pH 7.5) to a final concentration of 0.75 M at room temperature for 5 min, then on ice for 15 min. Cells were washed twice with 1X TDB and then crosslinked again by addition of freshly made DSG solution to the final concentration of 3 mM for 45 min at room temperature. The crosslinking reaction was quenched by addition of 2.5 M Tris buffer as described above. Cells were washed once in 1X TDB and snap-frozen in liquid nitrogen in aliquots of 50 million cells.

Frozen cells were resuspended in 200 µL of cold 1X TDB per 50 million cells. After 20 min incubation on ice, cells were collected by centrifugation (10,000 g, 5 min at 4 °C), washed once with 400 µL and once with 200 µL complete MB#1 buffer (10 mM Tris-HCl, pH 7.5, 50 mM NaCl, 5 mM MgCl2, 1 mM CaCl2, 0.2% NP-40, 1x Roche cOmplete EDTA-free (Roche diagnostics, 04693132001)) and resuspended in 50 µL of complete MB#1 buffer.

Chromatin was fragmented for 10 min at 37 °C with MNase concentrations pre-titrated to yield mostly mononucleosomal fragments (Merck, #N3755). The digestion was stopped by addition of 4 mM EGTA and incubation at 65 °C for 10 min. Here, chromatin aliquots were pooled for further processing. 200 million cell-equivalent per

replicate yielded the best results. Chromatin was collected by centrifugation (16,000 g, 10 min at 4 °C), washed with 500 μL of 1x NEBuffer 2.1 (NEB, #B7202S), collected by centrifugation (16,000 g, 10 min at 4 °C), and resuspended in 45 μL of NEBuffer 2.1. DNA ends were dephosphorylated by addition of 5 μL of rSAP (NEB, #M0203) and incubation at 37 °C for 45 min, then the reaction was stopped by incubation at 65 °C for 5 min. DNA overhangs were generated by adding 40 μL of "overhang mix" (5 μL of 10x NEBuffer 2, 0.2 μL of 200X BSA, 2 μL of 100 mM ATP (Thermo Fisher, #R0441), 3 μL of 100 mM DTT, 29.8 μL of $H_2O$, 8 μL of Large Klenow Fragment (NEB, #M0210L), 2 μL of T4 PNK (NEB, #M0201L)) and incubation at 37 °C for 15 min. The DNA overhangs were filled by adding 100 μL of "fill-in" mix (25 μL of 0.4 mM Biotin-dATP (Invitrogen, #19524-016), 25 μL of 0.4 mM Biotin-dCTP (Invitrogen, #19518-018), 1 μL of 10 mM dGTP, 1 μL of 10 mM dTTP, 10 μL of 10X T4 DNA Ligase Reaction Buffer (NEB #B0202S), 0.5 μL of 200X BSA (NEB, #B9000S), 37.5 μL of $H_2O$) and incubation at 23 °C for 3.5 h with interval shake. The reaction was stopped by addition of 12 μL of 0.5 M EDTA and incubation at 65 °C for 20 min.

Chromatin was collected by centrifugation (16,000 g, 10 min at 4 °C), washed with 1 mL of 1X T4 DNA Ligase Reaction Buffer (NEB #B0202S) and collected by centrifugation (16,000 g, 10 min at 4 °C). The chromatin pellet was resuspended in 1.8 mL of "proximity ligation" mix (1X T4 DNA Ligase Reaction Buffer (NEB #B0202S), 1X BSA (NEB, #B9000S), 11,200 U of NEB T4 Ligase (NEB, #M0202L)) and incubated rotating at room temperature for 2.5 h.

After proximity ligation, the chromatin was collected (>18,000 g, 10 min at 4 °C), resuspended in 200 μL of 1X of NEBuffer 1 (NEB, #B7001S) and 200 U of NEB Exonuclease III (NEB, #M0206S), and incubated for 5 min at 37 °C to remove biotin from unligated ends. Crosslinks were reversed by adding 31.25 μL of 20 mg/mL Proteinase K (NEB, P8107) and 25 μL of 10% SDS (65 °C, overnight).

The chromatin was phenol/chloroform purified and separated on a 1.5% agarose gel. Fragments between 250-400 bp in size were extracted and purified using the NucleoSpin Gel and PCR Clean-up kit (Macherey & Nagel) and resuspended in 56 μL of elution buffer. For end repair, 15 μL of end-repair mix was added (7 μL of 10X T4 DNA Ligase Reaction Buffer (NEB #B0202S), 1 μL of 25 mM dNTP mix, 2.5 μL of 10 U/μL NEB T4 PNK (NEB, M0201), 2.5 μL of 3 U/μL NEB T4 DNA polymerase I (NEB, M0203), 0.5 μL of 5 U/μL NEB DNA polymerase I, Large (Klenow) Fragment (NEB, M0210), 1.5 μL of $H_2O$) and incubated for 30 min at 20 °C and then for 20 min at 75 °C after addition of 0.5 M EDTA, pH 8 to a final concentration of 10 mM.

To isolate biotin-labelled ligation junctions, 50 μL of 10 mg/mL Dynabeads™ MyOne™ Streptavidin C1 beads (Invitrogen, #65001) were washed with 400 μL of 1X Tween washing buffer (TWB; 5 mM Tris-HCl pH 7.5, 0.5 mM EDTA, 1 M NaCl, 0.05% Tween 20), collected with a magnet, resuspended in 400 μL of 2X binding buffer (10 mM Tris-HCl pH 7.5, 1 mM EDTA, 2 M NaCl) and added to 330 μL of TLE (10 mM Tris-HCl (pH 7.5), 0.1 mM EDTA). 70 μL of the sample were added to the beads and incubated at room temperature for 15 min with slow rotation. The beads were washed once with 400 μL of 1X binding buffer, once with 100 μL of 1X TLE and finally resuspended in 41 μL of 1X TLE.

### Sequencing library preparation
For polyadenylation, 5 μL of 10X NEBuffer 2.1, 1 μL of 10 mM dATP and 3 μL of 5 U/ μL of Klenow fragment (3′→5′ exo-) (NEB, M0212) and incubated for 30 min at 37 °C followed by deactivation for 20 min at 65 °C. Beads were reclaimed with a magnet, washed once with 400 μL of 1X Quick ligation buffer (NEB, M2200) and resuspended in 46.5 μL of 1X Quick ligation buffer (NEB, M2200). 2.5 μL of DNA Quick ligase (NEB, M2200) and 0.5 μL of 50 μM annealed TruSeq adapters were added and incubated for 1 h at room temperature. Beads were separated on a magnet, resuspended in 400 μL of 1X TWB (5 mM Tris-HCl, 0.5 M EDTA, 1 M NaCl, 0.05% Tween-20) and washed for 5 min at room temperature with rotation. On the magnet, beads were washed twice

with 200 μL of 1X binding buffer and once with 200 μL of 1X NEBuffer 2.1 and resuspended in 20 μL of 1X NEBuffer 2.1. Each library was amplified in 12 separate reactions of 25 μL. Per reaction, 0.85 μL of 25 μM TruSeq PCR primer cocktail (TruSeq PCR primer cocktail_F: 5′-AATGATACGGCGACCACCGAG-3′; TruSeq PCR primer cocktail_R: 5′-CAAGCAGAAGACGGCATACGAG-3′), 12.5 μL of 2X Kapa HiFi HotStart Ready Mix (Kapa Biosystems, KR0370) and 10.15 μL of water were added to 1.5 μL of library bound to the beads. Amplification was performed as follows: 3 min at 95 °C, 8 cycles of 20 s at 98 °C, 30 s at 63 °C and 30 s at 72 °C, 1 cycle of 1 min at 72 °C, hold at 4 °C. The PCR reactions were pooled and the beads were removed from the supernatant using a magnet. Each library was purified by addition of 1.5 volumes of Agencourt AMPure XP beads (Beckman Coulter), according to the manufacturer's instructions. The sample was eluted off beads using 25 μL of 1x TLE buffer, transferred to a fresh tube and the concentration was determined using Qubit (Qubit dsDNA HS Assay Kit, Thermo Fisher) and qPCR (KAPA SYBR FAST qPCR Master Mix, Kapa Biosystems), according to the manufacturer's instructions. Library size distributions were determined on a 5% polyacrylamide gel. The samples were sequenced on an Illumina NextSeq 1000 or NextSeq 2000 on 60 base pair paired end mode.

### Chromosome conformation capture data processing and visualization
For visualization and downstream analysis, Micro-C datasets were processed according to the distiller pipeline (https://github.com/mirnylab/distiller-nf). Briefly, reads were mapped to our *T. brucei* Lister 427 version 12 (Tb427v12) genome assembly, working version. Mapping was done with bwa mem with flags -SP. Alignments were parsed and pairs were classified using the pairtools package[100] to generate 4DN-compliant pairs files. Pairs having matching alignment strands and coordinates with a possible 1 bp offset were considered PCR and/or optical duplicates and thus removed. Pairs classified as uniquely mapped or rescued chimeras with high mapping quality scores on both sides (MAPQ > 30) were corrected for ploidy as described below and then aggregated into contact matrices in the cooler format (cool files) using the cooler package[101] at 1 kb and into multiresolution cooler files (mcool files; resolutions: 1 kb, 2 kb, 5 kb, 10 kb, 50 kb, 100 kb).

Given the highly repetitive nature of the *T. brucei* genome, Micro-C datasets were in parallel processed with the mHi-C pipeline[102] to avoid underestimation of the biological signal from genomic repeats. The two ends of each paired read were mapped using bwa (v0.7.17[103]) with parameters set according to the 4DN guidelines. Re-ligation events, too short interactions and PCR duplicates were excluded. The genome was binned at 5 kb and 10 kb; uniquely mapping reads were binned as well, and multi-mapping reads were probabilistically allocated to a bin if the posterior probability of belonging to the bin was at least 0.6. Both uniquely and multi-mapping interactions were aggregated into contact matrices. Ploidy correction of the contact matrix was implemented as described below. All contact matrices were normalized using the iterative correction procedure[56] and converted to cool files, then extended to multi-resolution cool files (mcool files) and visualized using HiGlass[104].

### Ploidy normalization and conversion to entire chromosomes
To analyze sequencing data from the *T. brucei* chromosome core regions, where the two homologous chromosomes are very similar and the aligner cannot confidently assign a read to one allele or the other, we collapsed the two alleles into a single contig in the reference genome. In contrast, the subtelomeric regions of each homologous chromosome were kept separate. This resulted in the mapper assigning reads from two physical chromosomal regions to one core contig, leading to an overestimation of interaction frequencies in the core regions and a discrepancy between the true and measured interaction frequencies. To address this, the overestimation must be corrected

after alignment. Because similar issues arise when ploidy varies between chromosomes, we refer to this process as "ploidy correction".

A ploidy correction script based on Schmidt et al.[53] was implemented that 1) duplicates each interaction occurring in a pairs file and 2) splits the interactions coming from the homozygous cores into interactions coming from computationally defined coreA and coreB. Consequently, a computationally phased genome is defined. This way to correct for ploidy assumes that interactions occur preferentially within the same haplotype. Analogous scripts to convert interaction matrix and annotation files were implemented as well. After correcting for ploidy, the resulting files were bioinformatically processed to place interactions from connected contigs consecutively, so that a group of contigs that constitute a chromosome would be considered as such in all the following analyses of Micro-C data. As mentioned above, the working version of the Tb427v12 genome assembly has only one core region per chromosome (corresponding to core A). As a result, the analyses done on the bioinformatically-generated diploid genome (diploid_coreA genome) consider the annotated features of core A twice, while features of core B are not taken in consideration in downstream analyses.

### Reproducibility of Micro-C libraries

We used HiCRep[105] to do distance-corrected correlations of the various replicates. Correlation is calculated as follows: first, interaction maps are stratified by genomic distances and the correlation coefficients are calculated for each distance separately; afterwards, the reproducibility is determined by a stratum-adjusted correlation coefficient statistic (SCC) by aggregating stratum-specific correlation coefficients using a weighted average. As input files, we used genome-wide matrix files (bin1 bin2 value) at 100 kb resolution. The results are displayed as correlation heat map in Supplementary Fig. 5.

### Calling of enriched interactions

Enriched interactions outside the diagonal signal were called using Mustache[54]. At 2 and 5 kb resolution, such enriched contacts were called setting the *p*-value threshold parameter of 0.02 and the sparsity threshold parameter to 0.85. The program outputs the pairs of genomic coordinates representing called contacts and the number of contacts detected was derived from the number of entries in each output file.

### Pairwise TSSs enrichment

For each pair of TSSs, all normalized Micro-C matrix entries at the intersection of the pair were averaged. As control region, both annotations were shifted 20 kb downstream to compute a second average of entries as before. Enrichment was calculated by dividing each sample average by its control average. Each pair of TSSs was considered enriched if its average value was at least 1.25-fold the respective control average.

### Compartment analysis

Chromosomal compartment profiles were obtained using the FAN-C package[8] (fanc compartments function) through an eigenvector decomposition procedure on observed-over-expected cis contact maps (Micro-C, matrix obtained according to the distiller pipeline) at 10 kb resolution. Since in our Micro-C data the first eigenvector reflected other features of the heat map such as regions of no mappability (Supplementary Fig. 6), we calculated also the second eigenvector for compartment calling. Eigenvectors were oriented using H2A.Z MNase-ChIP-seq data[55] enrichment.

### Insulation score and domain callers

Insulation score was obtained using the FAN-C package[8] (fanc insulation function) and calculated at 2.5, 5 and 10 kb resolution on a Micro-C

matrix obtained according to the distiller pipeline. TAD-like structures were called using the hicFindTADs of the HiCExplorer package[106] at 250 bp resolution and exported in a bed file.

### Mappability calculation

Single end reads (read 1) from biological replicate 1, technical replicates 1-3 were aligned against the genome with bwa (v0.7.16[103]) and the mappability calculated as ratio between alignments with mapq ≥30 and alignments with mapq ≥ 0 using bamCompare (deeptools v3.5[107]).

### Detection of no visibility region

Micro-C data were IC normalized to eliminate factorizable biases. This procedure gives no visibility to regions that do not contain mapped reads and that have high noise (i.e., regions with the fewest number of contacts). The no visibility regions were annotated by extracting all balances with cooler[101] balance function and extract bins with no values assigned.

### Aggregate Analyses

The Aggregate Peak Analysis (APA) implemented in the GENOVA package[108] averages the signal surrounding a two-dimensional set of locations. The APA of detected enriched interactions was carried out using as input i) contact-objects loaded from the Micro-C and Hi-C mcool files; and ii) the pairs of coordinates of Mustache-called enriched interactions. When providing a list of multiple contact-objects, the GENOVA package also performs a differential analysis. Moreover, to distinguish between intra-chromosomal TSS enriched interactions and inter-chromosomal TSS enriched interactions, the APA was carried out on the Micro-C contact-object providing as two-dimensional sets of regions the pairwise combination of the coordinates of annotated TSSs where i) both elements of each pair belong to the same chromosome; or ii) both elements of each pair belong to different chromosomes. As control, the same coordinates were shifted of 20 kb downstream.

The Cross Spatial Chromatin Analysis (CSCAn) makes an averaged contact analysis for crosswise pairs between elements of a list of genomic location. The CSCAn was carried on lists of single and divergent TSSs and single and convergent TTSs. The results of all aggregate analyses were plotted using the visualise function from the same package.

Aggregated PTU matrices were generated by flipping the interactions of all relevant regions (intra- or inter-chromosomal PTUs) to the forward strand and scaling the regions to the same size. Each region is also extended by 10% of its size in both directions. Aggregated interactions were balanced via cooler with the '--ignore-diags' parameter set to zero.

### ChIP-seq data analysis

H2A.Z MNase-ChIP-seq data[55], SCC1 ChIP-seq data[48] and RNAPII ChIP-seq data[66] were aligned to our Tb427v12 genome assembly, working version, using bwa (v0.7.16[103]). The SCC1 ChIP sample and RNAPII ChIP sample were normalized to their respective inputs (ratio of ChIP to input reads). Aligned reads were ploidy normalized and contigs belonging to the same chromosome merged. After SAM to BAM conversion, the aligned reads were sorted, indexed and duplicates were removed using SAMtools version 1.8[97]. H2A.Z peaks were called using MACS2[98] with -q 0.01, --maxgap 500, --extsize 300, and broad settings. Peaks in centromeric regions were manually removed. SCC1 peaks were called with standard (narrow) settings and --extsize 200. Metaplots along PTUs and around loci of interests (single and divergent TSSs, single and convergent TTSs) were generated using deeptools v3.5[107] computeMatrix and plotProfile functions. A custom script was implemented to overlap called enriched interactions with annotated features.

## Virtual 4C analysis

The virtual 4C function implemented in the cooltools package[100,109] was used to extract all the interaction frequencies of a given viewpoint with the rest of the genome from the Micro-C mcool files. All virtual 4C analysis used as input mHiC matrices (retaining multi-mapping reads) at 5 kb resolutions (viewpoints: tRNA loci and TSSs) or at 10 kb (centromeric and rDNA loci). The tab-separated output files were plotted using the PyGenomeTracks program[110] either against the entire genome or against a subset group of genomic elements (e.g., one tRNA vs all tRNA loci; one rDNA vs all rDNA loci; one centromere vs all centromeric regions).

## Statistical peak analysis

Virtual 4 C enrichment $p$-values were computed using a two-sample Kolmogorov–Smirnov test (KS-test). For each genomic locus, a local and a global sample were collected, representing the local and global distributions. The local sample consists of all virtual 4C values within a window of the specific locus, while the global sample contains all other values. The window size was chosen according to the viewpoint size (100 or 500 kb for centromeres, 10 kb for rRNA regions, 5 or 10 kb for tRNA regions). The chromosome of the viewpoint was excluded from this analysis. The null hypothesis for the KS-test was that the local distribution is larger or equal to the global distribution. All values in the local sample were divided by 2 before applying the KS-test to check for a $\geq$ 2-fold enrichment.

## Fluorescence In Situ Hybridization (FISH)

For DNA FISH experiments, Digoxigenin- and Alexa Fluor 594-labelled DNA probes were generated by PCR using standard conditions with OneTaq polymerase (New England Biolabs), with a 1:2 ratio of ChromaTide™ Alexa Fluor™ 594-5-dUTP (Invitrogen) or Digoxigenin-11-dUTP (Roche) and dNTPs were used in the reaction. CIR147 repeats[83,84] were amplified from *T. brucei* Lister 427 genomic DNA (Supplementary Table 4). A smear of products with various sizes was generated but only fragments lower than 500 bp were gel extracted and purified with NucleoSpin Gel and PCR Clean-up kit (Macherey & Nagel). To generate Spliced Leader (SL) probes, we divided the SL region into 5 fragments of around 200 bp, then amplified from *T. brucei* Lister 427 genomic DNA (Supplementary Table 4). Probes were gel extracted and purified with NucleoSpin Gel and PCR Clean-up kit (Macherey & Nagel). After gel purification, probes were further purified by ethanol precipitation and resuspended in water.

For each assay, cells were fixed in 2% formaldehyde in media for 10 min at room temperature, washed three times with PBS and finally resuspended in PBS. The cells were attached to poly-l-lysine-treated coverslips, then permeabilized with 0.5% Triton X-100 in PBS for 15 min at room temperature, washed three times with PBS. Afterward, cells were incubated in 0.1 N HCl for 5 min, washed twice in PBS at room temperature, and then treated with 1 mg/mL RNAse A (Invitrogen) in PBS for 60 min at 37 ˚C. Lastly, cells were washed with PBS three times and incubated overnight in 2X saline-sodium citrate (SSC)/50% formamide/50 mM sodium phosphate at room temperature. Cells were then blocked with hybridization buffer (2X SSC, 5X Denhardt's solution, 50% Formamide, 50 mM Sodium phosphate buffer, 1 mM EDTA, 100 µg/ml salmon sperm DNA, 10% dextran sulfate, pH 7.5-8) for 60 min at 37 ˚C. After the hybridization buffer was removed from the sample, 10 µL of hybridization solution containing the 1 ng/µL probe were added and the coverslip sealed with fixogum (Marabu). Samples were denatured on a heat block at 75 °C for 1 min, followed by overnight incubation at 37 °C. After hybridization, coverslips were washed with three 10-min washes in 4X SSC, 2X SSC and 1X SSC at 50 °C. Samples were then washed three times for 10 min in PBS with 0.05% Tween. Alexa Fluor 594-labelled

samples were then mounted in Vectashield with DAPI. Digoxigenin-labelled samples were incubated with a mouse anti-digoxigenin antibody (Abcam; clone 21H8) diluted 1:10,000 in 1% BSA in PBS for 2 h at room temperature. After washing three times for 10 min in PBS with 0.05% Tween, digoxigenin-labelled samples were incubated for 1 h with a donkey anti-mouse Alexa Fluor 488 antibody (Invitrogen) diluted to 1:500 in 1% BSA. Samples were washed in PBS with 0.05% Tween and mounted in Vectashield with DAPI.

For imaging, a wide-field fluorescence Leica DMI8 microscope with Lumencor Spectra X light source and a HC PL APO 100×/1.47 OIL objective was used. Images were captured with a Leica DFC9000 GT sCMOS camera. The images were acquired as $z$ stacks (0.15 µm) and further deconvolved using the Deconwolf software[111].

## Reporting summary

Further information on research design is available in the Nature Portfolio Reporting Summary linked to this article.

## Data availability

High-throughput sequencing Micro-C data and ultra-long ONT reads generated in this study have been deposited in the European Nucleotide Archive under primary accession number PRJEB76933 [https://www.ebi.ac.uk/ena/browser/view/PRJEB76933]. The resulting genome assembly is available at https://zenodo.org/doi/10.5281/zenodo.12683395. Previously published ChIP-seq and Hi-C data that were used in this study are publicly available at the European Nucleotide Archive or through GEO Series under accession numbers GSE98061 [https://www.ncbi.nlm.nih.gov/geo/query/acc.cgi?acc=GSE98061] (H2A.Z MNase-ChIP-seq[55]), GSE100896 [http://www.ncbi.nlm.nih.gov/geo/query/acc.cgi?acc=GSE100896] (SCC1 ChIP-seq[48], samples GSM3357444 and GSM3357445), GSE150253 [https://www.ncbi.nlm.nih.gov/geo/query/acc.cgi?acc=GSE150253] (RNAPII ChIP-seq[66], samples GSM5381488 and GSM5381492), PRJEB35632 [https://www.ebi.ac.uk/ena/data/view/PRJEB35632] and GSE100896 [http://www.ncbi.nlm.nih.gov/geo/query/acc.cgi?acc=GSE100896] (Hi-C datasets[30,48], merged in this study).

## Code availability

The code for the generation and quality control of the genome assembly has been deposited at Zenodo https://zenodo.org/doi/10.5281/zenodo.12683395. The code for the analysis of Micro-C and Hi-C data is available at https://zenodo.org/doi/10.5281/zenodo.12683439.

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

## Acknowledgements
We thank N. Krietenstein for advice on the Micro-C protocol. We thank M. Oudelaar, I. Solovei and all members of the Siegel, Allshire, Ladurner, Meissner and Boshart laboratories for valuable discussions, T. Straub (Bioinformatics Core Facility, BMC) for providing server space, the Core Unit LAFUGA at the Gene Center Munich for next-generation sequencing and A. Thomae and S. Dietzel (Bioimaging Core Facility, BMC) for providing microscopes. This work was funded by the German Research Foundation (SI 1610 / 2-2 and 213249687—SFB 1064 to T.N.S.), an ERC Starting Grant (3D_Tryps 715466) and an ERC Consolidator Grant (SwitchDecoding 101044320) awarded to T.N.S. P.Y. was supported by an MSCA ITN Cell2Cell (86067) fellowship. The Medical Research Council (UK) supported this research through funding to R.C.A. and K.R.M. (MR/T04702X/1), along with Wellcome Trust funding to R.C.A. (106144 and 225237) and K.R.M. (221717) and to the Discovery Research Platform for Hidden Cell Biology (226791, Bioinformatics core).

## Author contributions
The experiments were designed by C.R. and T.N.S. and carried out by C.R. unless otherwise indicated. Genomic DNA extraction for ONT sequencing was carried out by A.B.S., R.C. and S.K. Assembly of the new genome reference and quality controls were performed by M.R.S. with the contribution of R.O.C. Annotation of the new reference genome was performed by M.R.S. and P.T. under the supervision of R.C.A. and K.R.M. Micro-C data were analyzed by C.R. and M.R.S. ChIP-seq data analysis was performed by C.R. and P.T. FISH experiments were performed and analyzed by P.Y. The work was supervised by T.N.S. The manuscript was written by C.R. and T.N.S. and edited by all co-authors. All authors contributed to the data interpretation and development of a model. Figures were generated by M.R.S. and C.R.

## Funding

## Competing interests
The authors declare no competing interests.
