## [Transparent Peer Review file · Nature Communications]

Inter-chromosomal transcription hubs shape the 3D genome architecture of African trypanosomes

Corresponding Author: Professor T. Nicolai Siegel

Version 0:

Reviewer comments:

Reviewer #1

(Remarks to the Author)

In this study, the authors investigate the 3D architecture of *Trypanosoma brucei* using Micro-C and report a more contiguous genome assembly using ultra-long sequencing. The availability of this assembly significantly contributes to parasite genomics, and this work provides novel insights into how the genome is organized within the nucleus. The authors focus on transcription hubs associated with polymerase classes I, II, and III to elucidate key aspects of nuclear organization and interchromosomal interactions.

It is important to note that, although not the primary focus of the study, the resequencing of the Lister 427 *T. brucei* genome allowed for the separation of haplotypes, which is a significant achievement. This excellent genome assembly provides a valuable resource for future research on these unicellular parasites.

The manuscript is well-written and effectively demonstrates the relevance of the results. However, several important aspects of the manuscript and analyses need to be carefully revised.

Major Comments:

1- The authors mention several times that *T. brucei* lacks TAD-like structures. However, their results show the presence of chromatin folding domains (CFDs) in *T. brucei*. For instance, chromosome 2 exhibits chromatin structures, a feature observed across all 11 chromosomes. To discard the presence of chromatin domains, Hi-C maps need to be analyzed using algorithms that predict them. Furthermore, there are observable differences between subtelomeric and core regions, indicating distinct organizational patterns in these areas.

It is important to note that, due to the relatively small size of *T. brucei* chromosomes compared to typical mammalian chromosomes, conventional mammalian-like TADs are not expected. Indeed, a single *T. brucei* chromosome can be comparable in size to the longest mammalian genes. In this context, TAD-like structures in *T. brucei* should be understood as CFDs. Moreover, a recent publication has identified chromatin folding domains (CFDs) in *T. brucei* (Díaz-Viraqué et al., doi: 10.1038/s41564-023-01483-y.), which contradicts the assertion that these organisms lack TAD-like structures.

Given these points, we strongly suggest that the authors revisit their claims in the Introduction, Results, and Discussion sections. The statement that *T. brucei* lacks TAD-like structures should be reconsidered in light of their findings and recent literature.

2- If chromosomal functional domains (CFDs) exist, it is difficult to assert the non-existence of chromosomal territories without substantial evidence. The authors claim that chromosomal territories do not exist, yet no supporting evidence is provided for this assertion. To substantiate this claim, a three-dimensional reconstruction study of the nucleus should be conducted to clearly demonstrate whether the chromatin organization resembles a "bowl of spaghetti" structure and to rule out territorial-like arrangements of the chromatin. We recommend that this point be reconsidered or removed from the manuscript until appropriate evidence is provided in future works.

3- The assertion that chromosomes in *T. brucei* fold into a Rab1-like organization is not clear. According to

Hoencamp et al. (doi: 10.1126/science.abe2218), there are four features of nuclear architecture at whole-chromosome scale: 1) Enhanced contact frequency between loci on the same chromosome, consistent with the existence of classical chromosome territories; 2) prominent contacts between centromeres; 3) Prominent contacts between telomeres; 4) An X-shape appears on the chromosomal map/an architecture in which the two arms of each chromosome fold back onto each other. Hoencamp et al. add: "The latter three features are reminiscent of the Rabl chromosome configuration, in which centromeres cluster and chromosome arms are arranged in parallel. We will call these features "Rabl-like".

The interaction maps generated in this work clearly show that the most frequent interactions are intrachromosomal, which would indicate that it is not a Rabl organization. We strongly suggest that the authors reconsider this statement. Perhaps *T. brucei* does not fall into the general classification of Hoencamp et al. and is a hybrid organization between the two models.

Minor Comments:

Since a highly relevant contribution of this paper is the deep sequencing of the genome and the use of ultra-long reads, where authors were able to distinguish haplotypes, they should highlight this by mentioning that it is a "phased genome".

When authors mention RNAPIII-transcribed genes, in addition to references 76-78, they should include Padilla-Mejía et al. (doi.org/10.1186/1471-2164-10-232)

Fig 3A first panel: include the features (TSS, etc.)

We propose to discuss the relevance of the intrachromosomal interactions with respect to the described interchromosomal interactions.

The authors assert that "transcription initiation of the PTUs is unregulated." However, this statement contradicts existing research, which has demonstrated that some mechanisms regulating transcription initiation do exist (10.1016/j.celrep.2021.110221). Then, this assertion should be revised.

(Remarks on code availability)

Reviewer #2

(Remarks to the Author)

(Remarks on code availability)

Reviewer #3

(Remarks to the Author)

Overall, this was an excellently written and careful study of the 3D organization of *T. brucei*. It was one of the more refreshing papers that I have reviewed, in its analysis and interpretation. I commend the authors on this work and provide the following recommendations to help them clarify a few key points.

Major Comments:

I could not get access to the data, and so could not complete a thorough review. The authors uploaded the data to GEO, but I did not find the reviewer access token anywhere in the manuscript or other materials. I was surprised because I thought this would be necessary before it was sent for review.

The authors state that the pattern is similar to compartments, but is it compartmental? The authors should call compartments using the eigenvector or CScore at high resolutions, to determine if the clustering of transcribed regions can be thought of as compartments (even in mammals, 500 bp -1 kb compartments were shown here: PMIDs: 37157000 & 37280210). The use of 50 kb eigenvector is far too coarse for compartments, especially if there are other overlaying features, as was shown in *C. elegans*: <https://www.nature.com/articles/s41588-024-01832-5>. In doing so, they could then test if the units form "discordant compartments", where the interaction signal falls off in the middle of paused genes (if they can classify units as paused or not). PMID: 37280210.

Fig 3A and 3B should also plot the enrichment signal of RNA Polymerase, to show whether it is fairly even across the unit, or builds up at TSSs, or builds up similar to cohesin. The results of this may change the interpretation regarding the role of cohesin (for example, if cohesin movement is limited by polymerase or not).

Fig 3B should also have a 2D scaled metaplot of Micro-C signal, to clearly demonstrate whether the units form domain like interactions, and whether the signal builds up coinciding with the build of cohesin.

They should include a table of Micro-C sequencing stats, including the number of reads as total sequenced, mapped, deduplicated, <20 kb, > 20kb, interchromosomal. It would also be valuable to provide some sort of metric of coverage. For example, due to the smaller genome size, the total number of contacts we obtained would be equivalent to X number of contacts in humans.

Minor comment:

Due to the topic and the lack of CTCF in *T. brucei*, I recommend that the authors use more specific language when discussing TADs, to avoid propagating the ambiguity surrounding the term. While it's true that many use the term TAD for loop extrusion domains / corner dot domains, they are not synonymous. TAD callers simply identify triangles near the diagonal, which several groups have shown in mammals are a mix of compartment domains, CTCF loop domains, and "ordinary" domains. See PMIDs: 28985562; 28826674; 37280210. In the introduction they should state that loop domains are flanked by CTCF (not that TADs are). Alternatively, if the authors wish to use the term TAD, I recommend they start by acknowledging the ambiguity, and that in their manuscript they are using the newer definition, outlined here: 31925403, to mean corner dot domains.

(Remarks on code availability)

The access to the data was not provided. The Zenodo links they provided were also not publicly available, and so I had no way of reviewing this information.

Reviewer #4

(Remarks to the Author)

The work by Rabuffo et al. presents advancements in the analysis of the 3D genome structure of *T. brucei*, highlighting inter-chromosomal interactions that form hubs involving centromeres and RNA-class-specific loci. The results are novel and intriguing, broadening our understanding of gene expression regulation mechanisms. Additionally, the study shows improvements in genome assembly using ultra-long reads, leading to the closure of many gaps. However, some conclusions, particularly regarding the formation of a "Rabl-like structure," are not well-supported by experimental evidence. Furthermore, certain findings (such as interactions between TSSs, tRNAs, and centromeres) were not thoroughly explored but could significantly enrich the analyses. Below, I outline some key points:

1. Regarding Genome Assembly: The authors claim that the *T. brucei* genome forms a "Rabl-like structure" (see my comments below). This type of genomic structure could complicate or hinder genome assembly when using Hi-C structural data (as previously done by the group). I understand that detecting a Rabl-like structure, which tends to have stronger interactions between centromeres and telomeres regardless of their linear genomic proximity, could lead to incorrect clustering of scaffolds around these regions. The authors should discuss or highlight the strategies used to minimize this bias.

Another issue that I find often overlooked in *T. brucei* genome articles is that only the assembly (and 3D structure) of megabase chromosomes is mentioned. The 3D structure of mid- and mini-chromosomes is not discussed, and the impact of excluding them from Micro-C or Hi-C analyses could potentially affect the results. This should be addressed.

2. Regarding Micro-C Processing Methodology: It's unclear to me why the authors "corrected our Micro-C data for ploidy by duplicating interactions from underrepresented regions⁵⁴." The rationale for mapping differences in core and subtelomeric regions, in different haplotypes, should be better explained. Is this approach also used for other organisms with significant haplotype differences like *T. brucei*?

8. Regarding TADs: The authors report the absence of TADs in *T. brucei* and cite their previous paper (Muller et al.). However, neither Muller nor the present study uses algorithms for TAD detection, but visual inspection of the Hi-C matrices in Muller et al. suggest the presence of TAD-like structures. How are the authors so certain that TADs do not exist in *T. brucei*? The absence of TADs in *T. brucei* also contrasts with their presence in phylogenetically related organisms like *T. cruzi*, as shown in Díaz-Viraqué et al e <https://doi.org/10.1101/2024.07.01.601582>.

3. Regarding RNA-Loci Specific Interactions: The authors should further explore (both quantitatively and functionally) their findings regarding these interactions, but they only superficially mention them. Below are some comments:

o Regarding TSS-TSS interactions: 1. What are the thresholds/statistical analyses used to obtain enriched interactions, as shown in Fig 2b? In Fig 2b, do the enriched interactions always originate from a TSS? Of the 367 enriched interactions, 233 are from TSSs, 7 from TTSSs, and 2 from tRNAs; what about the rest? 2. There is no information on the number of intra- or inter-chromosomal TSS-TSS interactions considered enriched. A graph or table showing this difference would be helpful. It would also be valuable to assess whether there is any functional association between PTUs that are or are not associated in 3D. Regarding the controls used in these analyses, why was a control containing sequences of equal size and GC content to the TSSs not used?

o Since *T. brucei* lacks transcriptional regulation, the authors could better discuss the biological impact of TSS clustering. Moreover, Pol II clustering in *T. brucei* was previously observed in Cordon-Obras et al. and should be cited. Does the number of Pol II clusters correspond to TSSs clusters?

o A quantitative analysis of the percentage of interaction between rRNA loci and tRNA loci would be beneficial. An aggregate plot summarizing the findings in Fig 4 would also be helpful. Again, it is necessary to report the criteria used to consider loci interactions, as well as the distance (in Kb, overlap?) considered significant for interaction. In this context, it would be important to zoom in on certain loci to better illustrate these interactions.

o Is there anything unique about the 25% of tRNAs that interact? Are they located in genomic regions that characterize new TSSs? Does the rRNA locus on chr8A interact with tRNAs since both are transcribed by Pol III? Do other genes transcribed by Pol III also interact with each other and with tRNA loci? The authors should explore this further.

o Create aggregate plots for centromeric, tRNA, and telomeric regions.

4. Regarding Centromeres: The authors mention in the text that centromere 4 clusters with centromere 5 (Fig Sup 8-10), yet they show via FISH that centromere 8 also interacts with them. However, centromere 8 does not interact with 4 and 5 in the micro-C data. How this can be explained? Again, what criteria were used to detect enrichment and colocalization in these virtual 4C assays? Other peaks could be detected in these 4C analysis. For example, centromere 8 appears to interact with centromere chr10 (Fig S8). Did the authors notice any peculiar interaction patterns between centromeres located in core and subtelomeric regions? Do core centromeres only interact with other core centromeres, and vice versa? What is the biological implication of this?

I would place figure S8 in place of Fig 5A, as the FISH validation refers to the interaction data shown in the supplement. For the FISH data, is CIR147 found only on chrs 4, 5, and 8? Why were these particular centromeres chosen? The authors mention different classes of repeats in the centromeres, now better resolved with the new sequencing. What are these classes? A table highlighting them would be important.

5. Regarding the Rabl-like Structure: In my opinion, this section requires more attention. The arguments and experiments leading to this conclusion are not robust. I don't believe the experiments convincingly demonstrate the existence of this type of organization in *T. brucei*.

o Is it possible to detect interactions among multiple centromeres in the micro-C matrices, as seen in yeast? The authors mention that centromeres are close to the nucleolus, showing FISH images. More images should be shown, along with nucleolar and nuclear membrane markers, simultaneously with telomere labeling, to illustrate the Rabl-like structure where centromeres and telomeres occupy opposite positions in the nucleus. Additionally, if centromeres are close to the nucleolus, was it possible to detect interactions between centromeres and rRNA loci by micro-C?

o The authors claim that *T. brucei* telomeres are clustered, as previously detected by Hi-C and FISH (Muller et al.). However, the Hi-C matrix (Extended Data Fig. 1B) and Micro-C data (from the current article) indicate that telomeres do not cluster. Therefore, *T. brucei* may possess a nuclear structure that differs from both Rabl-like and chromosome territories, independent of telomere clustering.

o Figure S6, which would also be important in suggesting the presence of a Rabl-like structure, is only mentioned in the discussion. However, the results are interesting and seem to conflict with the group's previously published results (Muller et al.), where they show more interaction between subtelomeric than core regions. Please discuss and explore this data.

o I believe the following sentence is incorrect: "Often, organisms with a Rabl-like configuration show a higher frequency of intra-chromosomal interactions compared to organisms whose chromosomes are organized into territories (Supplementary Fig. 12)." Additionally, it needs to be better explained what is meant by "the level of interactions between interchromosomal loci observed in *T. brucei* far exceeds that observed in other eukaryotes." What is the implication of this? Without a conclusion, the sentence feels incomplete.

6. Introduction: The introduction should be reformulated to improve its flow. Some sentences seem disconnected, or their linkage is unclear. For example:

o "Microscopy and Hi-C data from *T. brucei* have suggested that the 3D nuclear organization is important for RNA maturation^{31–33}. Here, highly expressed *T. brucei* genes were found to make frequent inter-chromosomal interactions with a locus that produces the noncoding spliced leader RNA that is trans-spliced to the 5' end of every mRNA and is thus crucial for mRNA maturation³⁴." Does "Here" refer to references 31-33 or the current article's data? If it refers to the current article, what are these highly expressed genes? rRNA?

o "Here, to investigate the role of inter-chromosomal hubs in the absence of chromosome territories, TADs, and E-P interactions, we generated high-resolution Micro-C contact maps in *T. brucei*." What is the reference for this? I understand that the lack of CTs has not been shown before and will be demonstrated in this paper; therefore, it should not be used in the introduction as a justification for using Micro-C.

Minor Comments:

- "In addition, 16 of the 21 annotated centromeres are now fully assembled, ranging in size from 9 to 55 kb" — shouldn't this be 22 centromeres?
- Figure 1A and other figures showing the genome (Fig 4, 5, and Supplementary): The colors showing the genomic features and gaps should be better highlighted for easier understanding. The black marking for "telomeric ends" is easily confused with dark blue, so are the centromeres.

(Remarks on code availability)

Reviewer #5

(Remarks to the Author)

This interesting manuscript uses state-of-the-art techniques (Micro-C) to reveal the 3D organization of the *Trypanosoma brucei* genome and show that there is considerable intra- and inter-chromosomal interaction between transcription start sites (TSSs) for the three different classes of RNA polymerase (RNAP). While this has previously been shown for RNAPI (which transcribes rRNA, as well as specific variant surface antigen genes), this now extends the phenomenon to the ~200 polycistronic transcription units (PTUs) of protein coding genes that are unique to this organism (and other Kinetoplastea), as well as (to a lesser extent) the tRNA and snRNA genes transcribed by RNAPIII. This finding highlights the evolutionary significance of inter-chromosomal interaction hubs for RNAP class-specific loci and centromeres and will likely be of considerable interest to the broader scientific community. In general, the manuscript is well-written and concise and is ready for publication with only minor revision, including the following:

1. There appears to be a minor error in Figure 2, where one of the PTUs (at the top of the figure) has arrowheads at both ends. This needs to be corrected.
2. It appears that the Micro-C experiments were carried out on only bloodstream (mammalian) forms of *T. brucei*. While it is probably not critical that they be repeated on procyclic (insect) forms, this should at least be mentioned in the Discussion.

(Remarks on code availability)

Version 1:

Reviewer comments:

Reviewer #1

(Remarks to the Author)

The manuscript has improved substantially, and all comments and suggestions have been addressed appropriately. Therefore, it is accepted for publication.

(Remarks on code availability)

Reviewer #2

(Remarks to the Author)

(Remarks on code availability)

Reviewer #3

(Remarks to the Author)

The authors responded to all of my comments and provide some new interesting analysis, particularly as it pertains to TSS interactions.

(Remarks on code availability)

Reviewer #4

(Remarks to the Author)

The authors have satisfactorily addressed my questions and those of the other reviewers. However, I still have three points that could be clarified.

1. Figure 3a: The new analyses using FAN-C at 10 kb resolution highlight the presence of three types of matrices whose compartmentalization is described as centromere-based, TSS-based, and core x subtelomeric-based. These analyses are indeed interesting; however, for me, in Matrix 2, it is possible to detect both centromere- and TSS-based compartmentalization. Additionally, I believe the matrix 3 does not show greater compaction in the subtelomeric regions but instead appears more compact in the core regions. Finally, in the first matrix, it seems evident that, in addition to centromere-based compartmentalization, TSS-based compartmentalization is also present.

Overall, my impression is that the type of compartmentalization is not exclusive. Does this pattern repeat in other chromosomes? I believe this should be mentioned in the text. To make this clearer, it might also be helpful to indicate in the matrices which of the three types of compartmentalization they correspond to.

2. The authors use two pipelines (distiller pipeline and mhic) to process and analyze the Micro-C data. However, it is not clear which pipeline was used to generate the input matrix for calculating the insulation score and identifying domains. This is particularly critical since the mhic pipeline leverages multimapped reads, which could impact mappability rates. In any case, the figure legends where mappability is shown (e.g., Fig. 3 and Supp. 7) should specify the origin of the matrix. Additionally, in Figure S7, it would be important to clarify what is meant by "no visibility biases."

3. The "3D clustering of tRNA loci" and "3D interactions between rRNA loci" have also been observed in a closely related species of *T. brucei*. Although these findings are described in a preprint (<https://www.biorxiv.org/content/10.1101/2024.07.01.601582v1>), both manuscripts share similar conclusions on these topics. This overlap suggests an evolutionary conservation between the two species, making it important to cite the preprint appropriately.

(Remarks on code availability)

Response to reviewers

We thank the reviewers for their constructive criticism and believe that the revised version of the manuscript addresses all the points raised by the reviewers. We have tried hard to better define the terminology used and apologize for any misunderstandings this may have caused. All relevant changes have been highlighted in the manuscript.

Reviewer #1 (Remarks to the Author):

In this study, the authors investigate the 3D architecture of *Trypanosoma brucei* using Micro-C and report a more contiguous genome assembly using ultra-long sequencing. The availability of this assembly significantly contributes to parasite genomics, and this work provides novel insights into how the genome is organized within the nucleus. The authors focus on transcription hubs associated with polymerase classes I, II, and III to elucidate key aspects of nuclear organization and interchromosomal interactions.

It is important to note that, although not the primary focus of the study, the resequencing of the Lister 427 *T. brucei* genome allowed for the separation of haplotypes, which is a significant achievement. This excellent genome assembly provides a valuable resource for future research on these unicellular parasites.

The manuscript is well-written and effectively demonstrates the relevance of the results. However, several important aspects of the manuscript and analyses need to be carefully revised.

Major Comments:

1- The authors mention several times that *T. brucei* lacks TAD-like structures. However, their results show the presence of chromatin folding domains (CFDs) in *T. brucei*. For instance, chromosome 2 exhibits chromatin structures, a feature observed across all 11 chromosomes. To discard the presence of chromatin domains, Hi-C maps need to be analyzed using algorithms that predict them. Furthermore, there are observable differences between subtelomeric and core regions, indicating distinct organizational patterns in these areas.

It is important to note that, due to the relatively small size of *T. brucei* chromosomes compared to typical mammalian chromosomes, conventional mammalian-like TADs are not expected. Indeed, a single *T. brucei* chromosome can be comparable in size to the longest mammalian genes. In this context, TAD-like structures in *T. brucei* should be understood as CFDs. Moreover, a recent publication has identified chromatin folding domains (CFDs) in *T. brucei* (Díaz-Viraqué et al., doi: 10.1038/s41564-023-01483-y.), which contradicts the assertion that these organisms lack TAD-like structures.

Given these points, we strongly suggest that the authors revisit their claims in the Introduction, Results, and Discussion sections. The statement that *T. brucei* lacks TAD-like structures should be reconsidered in light of their findings and recent literature.

We greatly appreciate the thoughtful feedback, which made us realize that we should have done a better job of defining the terms used to describe the many different contact domains revealed by the Hi-C data. We also apologize for not mentioning the very important *T. cruzi* Hi-C analysis published by Díaz-Viraqué et al.¹, Nature Microbiology, 2023 (PMID 37828247).

As suggested by ref 3, in the introduction we now acknowledge the ambiguity used to describe contact domains observed by Hi-C and related approaches and state that we adhere to the terminology suggested by Beagan & Phillips-Cremens², Nature Genetics, 2021 (PMID

31925403). Here, the authors use the term "contact domain" as an umbrella term to refer to any of the large set of self-associating chromatin domains described in the Hi-C literature. More specifically, they define:

chromatin domains as “Small triangles of enhanced contact frequency that tile the diagonal of each contact matrix”,

Compartment domains as: A chromatin domain whose boundaries align with inflection points in A/B compartmentalization signals

Loop as: A point of enriched contacts in Hi-C heat maps, appearing as a dot (a series of adjacent pixels with enhanced contact frequency with respect to the local chromatin domain structure)

Loop domain (TAD) as: A contact domain formed via loop-extrusion mechanisms

Compartments

We used FAN-C³ (Kruse et al., Genome Biology, 2020, PMID 33334380) to calculate the eigenvector typically used to identify compartment domains and found that subtelomeres often form distinct compartments from chromosome core regions, which we also clearly saw in our previous publications⁴ (Müller et al., Nature, 2018, PMID 30333624).

However, at 10 kb, the new Micro-C data also suggest in some cases a separation of chromosome arms into distinct compartments, and that sites of RNA pol II transcription initiation form a compartment separate from the rest of the core regions. Given that the entire core regions of chromosomes are transcribed, we did not expect to see core regions segregating into A and B compartments, but we were surprised to see that at 10 kb resolution, pol II transcription start sites appeared to segregate into a separate compartment. We suspect that this may be a result of clustering of RNA pol II TSSs.

TAD-like contact domains

We also used FAN-C³ and HiCEXplorer⁵ (Wolff et al. NAR, 2020, PMID 32301980) to calculate the insulation score, which is commonly used to identify TADs and other contact domains. However, the definition of TADs is somewhat more complicated than that of compartments and has changed over time. With our first Hi-C data generated in 2015 (published in 2018), we were advised by Job Dekker, a pioneer in the Hi-C field, not to refer to the observed contact domains as TADs, as the term TAD was used to describe contact domains in the range of a few 100kb. Smaller TADs were often referred to as TAD-like structures⁶ (Lajoie et al. Methods, 2015, PMID 25448293). However, since TADs are now defined² (Beagan & Phillips-Cremens, Nature Genetics, 2021, PMID 31925403) as contact domains caused by loop extrusion, and we have no evidence that the contact domains called by FAN-C or HiCEXplorer (based on the insulation score) are caused by loop extrusion, we don't think the regions flanked by high insulation scores should be called TAD or TAD-like.

Furthermore, while we also found that both FAN-C and HiCEXplorer frequently identified contact domain boundaries based on low insulation scores, similar to those shown in Díaz-Viraqué et al.¹, Nature Microbiology, 2023 (PMID 37828247), we also found that, with one exception, these sites of low insulation coincided with sites of low mappability.

Like our Hi-C for our previous analyses, our Micro-C data are IC normalized to give equal visibility to all genomic regions. However, there are several regions for which normalization does not work well due to low coverage caused by low mappability, repeats or errors in the genome. These regions are marked as 'no visibility'.

As we found that most sites with a low insulation score coincided with sites with no visibility, it is not clear whether the many contact domain boundaries reported by Fan-C and HiCEXplorer are simply reported due to low coverage or whether they represent bona fide contact domain boundaries.

Summary:

While we believe that the *T. brucei* genome is organized into compartments, the fact that almost all sites with low insulation scores coincide with low mappability makes it impossible to say whether they represent bona fide contact domain boundaries. However, we recognize that the absence of evidence does not prove the absence of TADs, and so we have corrected the manuscript to state that evidence for TAD-like structures is lacking.

We have now added a new section describing the results of the FAN-C eigenvector and insulation score calculations. We have also added the data from this analysis as a new Figure 3 and Supplementary Figures 6 and 7. In the Discussion section, we mention the intriguing possibility that regions of low mappability represent bona fide contact domain boundaries.

2- If chromosomal functional domains (CFDs) exist, it is difficult to assert the non-existence of chromosomal territories without substantial evidence. The authors claim that chromosomal territories do not exist, yet no supporting evidence is provided for this assertion. To substantiate this claim, a three-dimensional reconstruction study of the nucleus should be conducted to clearly demonstrate whether the chromatin organization resembles a "bowl of spaghetti" structure and to rule out territorial-like arrangements of the chromatin. We recommend that this point be reconsidered or removed from the manuscript until appropriate evidence is provided in future works.

We agree that it is difficult or impossible to assert the non-existence of a feature in biology. When we described the genome organization of *T. brucei* as Rabl-like, we had tried to follow the definition of Hoencamp et al.⁷ (Science, 2021, PMID 34045355). As we had interpreted the definition, any genome with observable centromere-centromere or telomere-telomere interactions should be called Rabl-like. In addition, our Hi-C data had revealed a relatively low ratio of intra- to inter-chromosomal interactions, which is reminiscent of other genomes thought to fold into a Rabl-like structure. However, after discussion with other experts in the field of genome architecture, we realized that centromere-centromere interactions are a very common phenomenon in biology, even for genomes of organisms with CTs, especially shortly after mitosis⁸ (Solovei and Mirny, Current Opinion in Cell Biology, 2024: PMID: 39180905). As suggested by reviewer 4, we believe that extensive DNA FISH experiments will be necessary to unambiguously describe the organization of the *T. brucei* megabase-sized chromosomes, which may also vary in organization throughout the interphase of the cell cycle. We believe that such experiments would be beyond the scope of this study and have therefore revised the manuscript as suggested by the reviewer.

In the abstract we now write:

“Our Micro-C data, analyzed in the context of this new genome assembly, reveal an intricate 3D genome organization. While we find that the genome has some features resembling chromosome territories, we also find that its chromosomes are organized around polymerase class-specific transcription hubs. RNAPI-transcribed genes interact with each other, as expected from their localization to the nucleolus, but we also found that RNAPII TSSs form inter-chromosomal transcription hubs with other RNAPII TSSs, highlighting the evolutionary significance of such inter-chromosomal hub structures”

3- The assertion that chromosomes in *T. brucei* fold into a Rabl-like organization is not clear. According to Hoencamp et al. (doi: 10.1126/science.abe2218), there are four features of nuclear architecture at whole-chromosome scale: 1) Enhanced contact frequency between loci on the same chromosome, consistent with the existence of classical chromosome

territories; 2) prominent contacts between centromeres; 3) Prominent contacts between telomeres; 4) An X-shape appears on the chromosomal map/an architecture in which the two arms of each chromosome fold back onto each other. Hoencamp et al. add: “The latter three features are reminiscent of the Rabl chromosome configuration, in which centromeres cluster and chromosome arms are arranged in parallel. We will call these features “Rabl-like”.

The interaction maps generated in this work clearly show that the most frequent interactions are intrachromosomal, which would indicate that it is not a Rabl organization. We strongly suggest that the authors reconsider this statement. Perhaps *T. brucei* does not fall into the general classification of Hoencamp et al. and is a hybrid organization between the two models.

As discussed in the response to the previous comment, we agree with the reviewer and have rewritten the manuscript accordingly.

Minor Comments:

Since a highly relevant contribution of this paper is the deep sequencing of the genome and the use of ultra-long reads, where authors were able to distinguish haplotypes, they should highlight this by mentioning that it is a “phased genome”.

We agree and are now mentioning this in the abstract. “In addition, to minimize artifacts from incorrectly scaffolded contigs, we used ultra-long sequencing reads to generate a highly contiguous phased genome assembly.” Lines 30-32.

When authors mention RNAPIII-transcribed genes, in addition to references 76-78, they should include Padilla-Mejía et al. (doi.org/10.1186/1471-2164-10-232)

We have included the suggested reference. Line 382.

Fig 3A first panel: include the features (TSS, etc.)

We have added the features to the panel, which is now Figure 4.

We propose to discuss the relevance of the intrachromosomal interactions with respect to the described interchromosomal interactions.

We have tried to clarify the relevance of these two types of interactions in the discussion by adding the following statement (lines 495-500):

“Taken together, our Micro-C data suggest that intra-chromosomal interactions leading to organization into compartments enclosing the transcribed core regions or the transcriptionally repressed subtelomeric regions provide the global genome organization. In contrast, inter-chromosomal interactions provide a much finer additional level of interactions between specific loci, such as RNAPII TSSs, RNA genes, or between the VSG and the SL array, that may be important for RNA transcription and processing.”

The authors assert that "transcription initiation of the PTUs is unregulated." However, this statement contradicts existing research, which has demonstrated that some mechanisms regulating transcription initiation do exist (10.1016/j.celrep.2021.110221). Then, this assertion should be revised.

The study by Cordon-Obras et al.⁹ (Cell Reports, 2021, PMID 35021094) identified promoter-like sequence elements and then, to test their potential to drive transcription, cloned these sequences upstream of a luciferase gene in a plasmid. After transient transfections, they found that the identified sequences could indeed drive transcription. In addition, they found that in the absence of chromatin, different plasmids containing different promoters resulted in different luciferase activities. From this, they concluded that transcriptional regulation may occur. However, stable transfection of the same plasmid into different genomic loci gave highly variable results, indicating that local chromatin structure and the degree of DNA accessibility are the main factors controlling transcription initiation. These observations, together with ChIP-seq data showing that all identified RNA Pol II transcription start sites are marked by the same chromatin structures (histone variants and histone modifications), suggest a lack of transcriptional control. Consistent with this assumption, our group recently measured nascent transcription using RNA labeling followed by transient transcriptome sequencing (TT-seq)¹⁰ (Luzak et al. BioRxiv, doi.org/10.1101/2024.06.18.599538) and found that PTUs are transcribed at remarkably similar levels, suggesting no PTU-specific regulation.

Thus, we believe that the present data do indeed support the notion that RNA pol II transcription is not regulated in *T. brucei*.

Reviewer #2 (Remarks to the Author):

Reviewer #3 (Remarks to the Author):

Overall, this was an excellently written and careful study of the 3D organization of *T. brucei*. It was one of the more refreshing papers that I have reviewed, in its analysis and interpretation. I commend the authors on this work and provide the following recommendations to help them clarify a few key points.

Major Comments:

I could not get access to the data, and so could not complete a thorough review. The authors uploaded the data to GEO, but I did not find the reviewer access token anywhere in the manuscript or other materials. I was surprised because I thought this would be necessary before it was sent for review.

We are very sorry to read that you have not been able to access the data. The raw data were uploaded to ENA and the processed data and codes were uploaded to Zenodo, both datasets should be fully accessible without any protection, i.e. no tokens should be needed. For ENA we had only listed the study accession number, for easier access we have now added the full hyperlink: <https://www.ebi.ac.uk/ena/browser/view/PRJEB76933>

The authors state that the pattern is similar to compartments, but is it compartmental? The authors should call compartments using the eigenvector or CScore at high resolutions, to determine if the clustering of transcribed regions can be thought of as compartments (even in mammals, 500 bp -1 kb compartments were shown here: PMIDs: 37157000 & 37280210). The use of 50 kb eigenvector is far too coarse for compartments, especially if there are other overlaying features, as was shown in *C. elegans*: <https://www.nature.com/articles/s41588-024->

01832-5. In doing so, they could then test if the units form “discordant compartments”, where the interaction signal falls off in the middle of paused genes (if they can classify units as paused or not). PMID: 37280210.

As suggested, we repeated our compartmental analysis and used FAN-C³ to determine eigenvector values at different resolutions. While the 1 kb resolution did not seem to yield any useful information regarding compartments (Supplementary Figure 6), the 10 kb resolution appeared to be more appropriate than the 50 kb resolution we had originally used. Therefore, we used FAN-C to determine the eigenvector values for all chromosomes at 10 kb and added the new data to the manuscript as Figure 3a and Supplementary Figure 6. The data suggest compartmentalization at three levels: 1) core vs. subtelomere, 2) a separation of chromosome arms 3) sites of RNA pol II transcription initiation vs. the remainder of the core regions.

Given that the entire core regions of chromosomes are transcribed, we did not expect to see core regions segregating into A and B compartments, but we were surprised to see that at 10 kb resolution, pol II transcription start sites appeared to segregate into a separate compartment. We suspect that this may be a result of clustering of RNA pol II TSSs.

Fig 3A and 3B should also plot the enrichment signal of RNA Polymerase, to show whether it is fairly even across the unit, or builds up at TSSs, or builds up similar to cohesin. The results of this may change the interpretation regarding the role of cohesin (for example, if cohesin movement is limited by polymerase or not).

We have plotted the distribution of RNA pol II along the PTU and find that it is enriched at the TSS. We have added the new plots to the figures in the manuscripts as suggested (now Figure 4 and 5a). As we had previously published^{11,12} (Wedel et al., EMBO J, 2017, PMID 28701485; Kraus et al., Nature Communications, 2020, PMID 32198348) and as has been reported for other organisms, we don't see RNA pol II enriched at transcription termination sites, but just after transcription initiation sites, suggesting transcriptional pausing. We don't think this distribution is inconsistent with a role for RNA pol II in 'pushing' cohesin to transcription termination sites. It seems plausible to us that at some point RNA pol II dissociates from DNA while cohesin remains in place.

Fig 3B should also have a 2D scaled metaplot of Micro-C signal, to clearly demonstrate whether the units form domain like interactions, and whether the signal builds up coinciding with the build of cohesin.

We have generated a 2D scaled metaplot of PTUs and included it in Figure 3 (now Figure 5b), for details see Methods. What we see is that TSSs interact with other TSSs and that cohesin is enriched at TSSs.

They should include a table of Micro-C sequencing stats, including the number of reads as total sequenced, mapped, deduplicated, <20 kb, > 20kb, interchromosomal. It would also be valuable to provide some sort of metric of coverage. For example, due to the smaller genome size, the total number of contacts we obtained would be equivalent to X number of contacts in humans.

We have included a table with the relevant information, Supplementary Table 2. Excluding duplicates, we were able to map >170 million reads; given the *T. brucei* genome size of 35 Mb

and the human genome size of 3.3 Gb, the resolution of our study should be comparable to a study with ~16 billion reads on the human genome.

We have added this statement to the Discussion section, lines 459-462:

“We then performed a high-resolution Micro-C analysis of bloodstream *T. brucei*, generating >170 million uniquely mapping reads. Given the *T. brucei* genome size of 35 Mb and the human genome size of 3.3 Gb, the resolution of our study is comparable to a study with ~16 billion reads of the human genome.”

Minor comment:

Due to the topic and the lack of CTCF in *T. brucei*, I recommend that the authors use more specific language when discussing TADs, to avoid propagating the ambiguity surrounding the term. While it's true that many use the term TAD for loop extrusion domains / corner dot domains, they are not synonymous. TAD callers simply identify triangles near the diagonal, which several groups have shown in mammals are a mix of compartment domains, CTCF loop domains, and “ordinary” domains. See PMIDs: 28985562; 28826674; 37280210. In the introduction they should state that loop domains are flanked by CTCF (not that TADs are). Alternatively, if the authors wish to use the term TAD, I recommend they start by acknowledging the ambiguity, and that in their manuscript they are using the newer definition, outlined here: 31925403, to mean corner dot domains.

We agreed that it is important to use very specific language to avoid ambiguity. We realize that we have caused unnecessary confusion by not defining our terminology more clearly, and we apologize.

In the Introduction, we now write: "To reduce ambiguity, in our study we follow the definitions proposed by Beagan and Phillips-Cremins in 2020 ..." and then give a summary of how the different contact domains are defined.

Reviewer #3 (Remarks on code availability):

The access to the data was not provided. The Zenodo links they provided were also not publicly available, and so I had no way of reviewing this information.

We are very sorry to read that you have not been able to access the data. The raw data have been uploaded to ENA and the processed data and codes have been uploaded to Zenodo, both datasets should be fully accessible without any protection, i.e. no tokens should be required. The two Zenodo links, one for the genome assembly and one for the Micro-C data analysis, are as follows

Genome Assembly:

<https://zenodo.org/doi/10.5281/zenodo.12683395>

Micro-C data analysis and processed data

<https://zenodo.org/doi/10.5281/zenodo.12683439>

To our knowledge, these are publicly available.

Reviewer #4 (Remarks to the Author):

The work by Rabuffo et al. presents advancements in the analysis of the 3D genome structure of *T. brucei*, highlighting inter-chromosomal interactions that form hubs involving centromeres and RNA-class-specific loci. The results are novel and intriguing, broadening our understanding of gene expression regulation mechanisms. Additionally, the study shows improvements in genome assembly using ultra-long reads, leading to the closure of many gaps. However, some conclusions, particularly regarding the formation of a "Rabl-like structure," are not well-supported by experimental evidence. Furthermore, certain findings (such as interactions between TSSs, tRNAs, and centromeres) were not thoroughly explored but could significantly enrich the analyses. Below, I outline some key points:

1. Regarding Genome Assembly: The authors claim that the *T. brucei* genome forms a "Rabl-like structure" (see my comments below). This type of genomic structure could complicate or hinder genome assembly when using Hi-C structural data (as previously done by the group). I understand that detecting a Rabl-like structure, which tends to have stronger interactions between centromeres and telomeres regardless of their linear genomic proximity, could lead to incorrect clustering of scaffolds around these regions. The authors should discuss or highlight the strategies used to minimize this bias.

For the reasons outlined below, we had described the *T. brucei* genome organization as Rabl-like in our previous submission. However, we now believe that this may not be the most appropriate characterization.

Nevertheless, we do not believe that a Rabl-like chromosomal organization would preclude the use of Hi-C data to scaffold its contigs. Typically, the first step in a de novo genome assembly is to generate contigs from long-read sequencing data. If chromosome conformation data, such as Hi-C, are available, contigs can then be scaffolded based on the frequency of DNA-DNA interactions. While centromere clustering can lead to a relatively high frequency of interactions between different chromosomes, the dynamic nature of these regions contrasts with the consistent proximity of loci on the same chromosome, resulting in more frequent interactions between proximal loci. Thus, contigs from the same chromosome are more likely to interact than those from different chromosomes.

The Hoencamp paper (Hoencamp et al.⁷, Science, 2021, PMID 34045355) describes the successful scaffolding of 24 genomes using Hi-C data, many of which exhibit Rabl-like organization. Similarly, the DNA Zoo project has used Hi-C-based scaffolding for many genomes, regardless of their specific genome organization. Therefore, although the ratio of intra- to inter-chromosomal interactions may be higher in genomes with chromosome territories than in those with a Rabl-like organization, short-range intrachromosomal interactions are still more common than frequent inter-chromosomal interactions due to the linear nature of DNA. Thus, Hi-C data should be helpful in scaffolding contigs even for genomes with a Rabl-like organization.

Because the use of Hi-C data for scaffolding and phasing of *T. brucei* data has been described elsewhere^{4,13} (Müller et al., Nature, 2018, PMID 30333624; Cosentino et al., Nucleic Acid Research, 2021, PMID 34541528), and we only use ONT data to fill gaps and correct repeat length in this study, we do not believe it is necessary to discuss the use of Hi-C data in scaffolding in this manuscript.

Another issue that I find often overlooked in *T. brucei* genome articles is that only the assembly (and 3D structure) of megabase chromosomes is mentioned. The 3D structure of mid- and mini-chromosomes is not discussed, and the impact of excluding them from Micro-C or Hi-C analyses could potentially affect the results. This should be addressed.

This is a very valid point and something we have been wondering about for years. However, until we have a high-quality scaffold of the intermediate and minichromosomes, it will be difficult to assess their impact on the overall *T. brucei* genome architecture. In addition, even when these chromosomes are fully scaffolded, their many repeat regions will likely make it difficult to analyze Hi-C reads from these chromosomes, leaving their genome organization open. We mention this point now in the discussion, lines 524-529:

“Furthermore, in this and previous studies, we have focused on the diploid set of the 11 megabase-sized chromosomes. However, *T. brucei* also contains 5-6 intermediate chromosomes and ~100 minichromosomes. Because the intermediate and minichromosomes are not yet fully assembled, they have been excluded from genome architecture analyses. Whether and how this affects genome architecture analyses remains to be determined in future studies.”

2. Regarding Micro-C Processing Methodology: It's unclear to me why the authors "corrected our Micro-C data for ploidy by duplicating interactions from underrepresented regions⁵⁴." The rationale for mapping differences in core and subtelomeric regions, in different haplotypes, should be better explained. Is this approach also used for other organisms with significant haplotype differences like *T. brucei*?

Since the publication of our first *T. brucei* Hi-C data along with the partially phased assembly of the *T. brucei* genome, in which the megabase chromosomes contained homozygous core regions and phased heterozygous subtelomeric regions⁴ (Müller et al., Nature, 2018, PMID 30333624), we have consulted with experts in the field on how best to analyze Hi-C data for this genome.

To analyze sequencing data from the core regions, where the two homologous chromosomes are very similar and the aligner cannot confidently assign a read to one allele or the other, we collapsed the two alleles into a single contig in the reference genome. In contrast, the subtelomeric regions of each homologous chromosome were kept separate. This resulted in the mapper assigning reads from two physical chromosomal regions to one core contig, leading to an overestimation of interaction frequencies in the core regions and a discrepancy between the true and measured interaction frequencies. To address this, the overestimation must be corrected after alignment. Because similar issues arise when ploidy varies between chromosomes, we refer to this process as "ploidy correction".

In previous studies, we addressed this discrepancy by halving the interaction frequencies of the bins in the core regions. Using this normalization approach, our Hi-C heatmaps suggested that the heterozygous subtelomeric regions were significantly more compact than the chromosome cores. More recently, after discussions with Anton Goloborodko (IMBA), a pioneer in Hi-C data analysis and normalization, we concluded that a more accurate method would be to take the interaction file (pairs file), duplicate each interaction, and then split the core-region interactions into those from bioinformatically defined CoreA and CoreB. This approach relies on two assumptions:

- Core interactions occur only between A-A or B-B regions.
- Core A always interacts with A arms (3' or 5').

The new ploidy normalization approach is described in detail in Schmidt et al.¹⁴ (Nucleic Acid Research, 2024 PMID 38281191), and we believe it provides a more accurate reflection of the true genomic compaction of the two region types. While we recognize that the underlying assumptions may not always hold, this method more accurately captures the interaction frequencies within each haplotype and better explains certain patterns in the nuclear

organization data, such as the distance-dependent decay of interactions, particularly between subtelomeric regions and the core of the same chromosome. Therefore, we applied this normalization to our current study. Although our Hi-C and Micro-C maps still show that subtelomeres are more compact than chromosome core regions, the difference in interaction frequency is now less pronounced than in previous heat maps published from our lab.

We have extended the Methods section to explain why we perform a ‘ploidy’ correction.

8. Regarding TADs: The authors report the absence of TADs in *T. brucei* and cite their previous paper (Muller et al.). However, neither Muller nor the present study uses algorithms for TAD detection, but visual inspection of the Hi-C matrices in Muller et al. suggest the presence of TAD-like structures. How are the authors so certain that TADs do not exist in *T. brucei*? The absence of TADs in *T. brucei* also contrasts with their presence in phylogenetically related organisms like *T. cruzi*, as shown in Díaz-Viraqué et al <https://doi.org/10.1101/2024.07.01.601582>.

As already discussed in our responses to reviewers 1 and 3, we apologize for not clearly defining the terminology used to describe the different contact domains.

As suggested by reviewer 3, we now acknowledge in the Introduction the ambiguity of the terminology used to describe contact domains observed by Hi-C and related approaches and state that we adhere to the terminology proposed by Beagan & Phillips-Cremins², Nature Genetics, 2021 (PMID 31925403). Here, the authors use the term "contact domain" as an umbrella term to refer to any of the large set of self-associating chromatin domains described in the Hi-C literature. More specifically, they define:

chromatin domains as “Small triangles of enhanced contact frequency that tile the diagonal of each contact matrix”,

Compartment domains as: A chromatin domain whose boundaries align with inflection points in A/B compartmentalization signals

Loop as: A point of enriched contacts in Hi-C heat maps, appearing as a dot (a series of adjacent pixels with enhanced contact frequency with respect to the local chromatin domain structure)

Loop domain (TAD) as: A contact domain formed via loop-extrusion mechanisms

Originally, TADs were defined to be in the 100kb range, and the term TAD-like structures was used to describe similar but smaller contact domains.

The reason why we have refrained from calling some of the apparent contact domains TAD-like is twofold.

- 1) We had noticed that most of the apparent contact domain boundaries coincided with regions of low mappability.
- 2) Based on the new definition of TADs, only contact domains formed by loop extrusion should be called TADs, and we have no evidence that the apparent contact domains are formed by loop extrusion.

To provide a little more context: TADs are typically called based on a large change in insulation score, but as Reviewer 3 notes, "TAD callers" simply identify triangles of increased contact frequency near the diagonal flanked by regions of high insulation, which may be a mixture of different contact domains. In addition, repetitive regions that cause low mappability and gaps in the assembly can result in low insulation scores and thus may be misidentified as TAD boundaries.

While we find that so-called TAD callers, e.g. used FAN-C³ (Kruse et al., *Genome Biology*, 2020, PMID 33334380) and HiCEXplorer⁵ (Wolff et al. *NAR*, 2020, PMID 32301980), frequently identify contact domains based on insulation profiles, we observed that almost all of the domain boundaries (sites of low insulation score) represent sites of low mappability caused by repetitiveness or incomplete genome assembly, thus the insulation sites may simply be an analytical artifact.

We have now added a new section to the manuscript in which we describe the results of the FAN-C insulation score calculations. We have also added the data from this analysis as a new Figure 3b and Supplementary Figures 7. In the Discussion section, we mention the intriguing possibility that regions of low mappability represent contact domain boundaries.

For more details, see also our response to reviewer 1 point 1 and reviewer 3.

3. Regarding RNA-Loci Specific Interactions: The authors should further explore (both quantitatively and functionally) their findings regarding these interactions, but they only superficially mention them.

Below are some comments:

o Regarding TSS-TSS interactions: 1. What are the thresholds/statistical analyses used to obtain enriched interactions, as shown in Fig 2b?

The enriched interactions were identified using the loop caller Mustache¹⁵ (Roayaei Ardakany et al., *Genome Biology*, 2020, PMID 32998764) with a p-value threshold parameter of 0.02 and a sparsity threshold parameter (used to filter out loops in sparse regions) of 0.85. This information was provided in the Methods section, but we have now added the name of the loop caller and the parameters to the legend of Figure 2.

In Fig 2b, do the enriched interactions always originate from a TSS?

Of the 367 enriched interactions, 233 are from TSSs, 7 from TTSSs, and 2 from tRNAs; what about the rest? 2.

We are sorry that the description of our analysis was not very clear and have now expanded the relevant section and the legend of Figure 2 to better describe the type of analyses we performed.

Using Mustache, we initially identified 367 enriched interactions (interaction between two bins). Of these 367 interacting bins, 233 are between pairs where both bins contain a TSS. 7 of these 233 interacting pairs contain both annotated TSSs and annotated TTSSs. This is not surprising as these sites are often very close to each other. 2 of the 233 interacting bin pairs contain annotated TSSs and also annotated tRNA genes. However, we don't think this is very relevant and so we have removed this panel from Figure 2.

We also examined the remaining 134 interaction pairs, those pairs where one or two of the interacting bins had no annotated TSS, and found that all of them fell into one of the following categories: i) one or both of the bins had low coverage, leading to high noise; ii) one bin had an annotated TSS and the other bin was immediately adjacent to a bin with an annotated TSS; iii) in three cases, one bin was not annotated to contain a TSS but had high levels of H2A.Z, so we assume that there was simply an annotation error. For resubmission, we corrected the annotation and adjusted the numbers; accordingly, iv) we noticed an error in the TSS annotation for chromosome 9. Here the annotation was shifted by ~15kb. For the resubmission, we corrected the annotation and adjusted the numbers accordingly.

There is no information on the number of intra- or inter-chromosomal TSS-TSS interactions considered enriched. A graph or table showing this difference would be helpful. It would also be valuable to assess whether there is any functional association between PTUs that are or are not associated in 3D.

To evaluate the enrichment of TSS interactions over their controls, we calculated the average interaction frequency between bins overlapping TSSs for each pair of TSSs and used as control regions the average interaction frequency of the same bins shifted 20kb downstream. TSS enrichment was calculated by dividing the sample average by the control average. TSSs and controls with no data (because of normalization or unmappability of the region) were excluded. Ratios were expressed as log₂ fold:

Of the 1,914 TSS-TSS interactions pairs for which interaction frequencies could be determined, 64.8% had a ≥ 1.25 -fold higher interaction frequency than their corresponding control region.

Of the 14,412 TSS-TSS inter-chromosomal interactions pairs for which interaction frequencies could be determined, 76.8% had a ≥ 1.25 -fold higher interaction frequency than their corresponding control region.

We have now added these numbers to the manuscript (lines 215-223) and expanded the Methods section to describe how the enrichment was calculated.

In addition, we investigated whether there is a functional association between PTUs that are associated or not associated in 3D. For this purpose, we overlapped their annotation with the Micro-C heatmap. However, we did not identify any specific patterns. The vast majority of "lowly interacting" TSSs were between TSSs located on different chromosomes and therefore showed a weak increase in interaction frequency. In addition, many of the lowly interacting TSSs and/or their controls were located in regions of high noise. Finally, we identified a few cases where the small size of the PTU meant that the control region 20 kb downstream was again in close proximity to a TSS.

Regarding the controls used in these analyses, why was a control containing sequences of equal size and GC content to the TSSs not used?

We used control regions of the same size as the TSSs. We felt that it was more important to have a large variety of control regions from different regions of the genome than to have them match in GC. Nevertheless, the average GC content of our control regions was very similar to that of the TSSs.

TSSs	Controls
AVERAGE %GC	AVERAGE %GC
46,58	47,40
STDEV %GC	STDEV %GC
2,357084	3,926242

o Since *T. brucei* lacks transcriptional regulation, the authors could better discuss the biological impact of TSS clustering. Moreover, Pol II clustering in *T. brucei* was previously observed in Cordon-Obras et al. and should be cited.

Yes, we should have cited that study showing RNAPII clustering and corrected this oversight. The study by Cordon-Obras et al.⁹ is now cited and discussed in the Discussion, line 490.

Does the number of Pol II clusters correspond to TSSs clusters?

The number of clusters can only be determined by imaging approaches. Hi-C and Micro-C do not capture multivalent interactions and only measure interactions at the population level. Therefore, Micro-C data do not provide information on how many sites are involved in a cluster at any given time.

Neither Cordon-Obras et al. nor we quantified the number of RNAP II and TSS loci when we imaged loci of H4K10ac, a histone mark enriched at TSS¹⁶ (Siegel et al., Genes & Development, 2009, PMID 19369410). Thus, we cannot say whether the numbers match and whether, for example, there are sites of RNAP II enrichment that are not associated with TSS.

o A quantitative analysis of the percentage of interaction between rRNA loci and tRNA loci would be beneficial. An aggregate plot summarizing the findings in Fig 4 would also be helpful. Again, it is necessary to report the criteria used to consider loci interactions, as well as the distance (in Kb, overlap?) considered significant for interaction. In this context, it would be important to zoom in on certain loci to better illustrate these interactions.

Virtual 4C peaks are influenced by a number of factors, including the size of the viewpoint, the size of the target, and the repetitiveness of both regions, making it difficult to identify "significant" peaks. We are not aware of any tools that have been developed to evaluate the significance of 4C peaks.

Therefore, we analyzed our 4C analysis qualitatively. To identify meaningful peaks, we considered whether their shape followed a profile expected from a distance-dependent decay of the interaction. Next, we evaluated whether the same high interaction was visible when the virtual4C was run using the target locus as the viewpoint.

While we don't believe that quantitative evaluation always improves the interpretation of biological data, we implemented a statistical method that allowed us to identify significant peaks. A description of how to identify significant peaks has been added to the Methods and significant peaks are now marked with red lines.

o Is there anything unique about the 25% of tRNAs that interact? Are they located in genomic regions that characterize new TSSs? Does the rRNA locus on chr8A interact with tRNAs since both are transcribed by Pol III? Do other genes transcribed by Pol III also interact with each other and with tRNA loci? The authors should explore this further.

As suggested, we evaluated the tRNA involved in the interactions and considered the following: type of amino acid binding site, transcript level, and whether they were adjacent to snRNA or snoRNA transcribed by RNAPIII. However, no pattern emerged.

No, the rRNA locus on chromosome 8A does not interact with any tRNA or RNAPIII transcribed gene.

Yes, we see snRNA and snoRNA transcribed by RNAPIII to interact with other tRNA genes. However, since snRNA and snoRNA are located adjacent to tRNA loci, it is difficult to say how relevant the observed interactions are.

We are not sure what the reviewer means with "regions that characterize new TSSs"

o Create aggregate plots for centromeric, tRNA, and telomeric regions.

As suggested, we have generated aggregate plots for centromeric, tRNA, and telomeric regions, see below. However, we don't believe that this analysis adds important insight to the manuscript and would prefer to leave it out.

Figure 1R. Aggregate Chromosome Analysis. Each chromosome arm is rescaled to a uniform length and the signal from all intra- and inter-chromosomal contacts are aggregated. This analysis aggregates the signal around centromeres and telomeres/subtelomeric regions and was based on Hoencamp et al.⁷ (Science, 2021, PMID 34045355).

Figure 2R. Aggregate Region Analysis (ARA; GENOVA package¹⁷ (van der Weide, Genes & Development, 2021, PMID 34046591) of tRNA annotated genes (n = 137).

4. Regarding Centromeres: The authors mention in the text that centromere 4 clusters with centromere 5 (Fig Sup 8-10), yet they show via FISH that centromere 8 also interacts with them. However, centromere 8 does not interact with 4 and 5 in the micro-C data. How this can be explained?

Our FISH data in Figure 5 (now Figure 7) visualizes the CIR147 repeat found at the centromeres of chromosomes 4, 5 and 8. Since most (>85%) of the cells show more than one CIR147 signal, we cannot say that the centromere of chromosome 8 colocalizes with the other two centromeres. All we can say is that there is centromere clustering. Looking at our Virtual 4C data, we see weak interactions between the centromere of chr 8 and those of chr 4 and 5. However, the newly implemented quantification approach does not consider them significant.

Thus, we don't see any discrepancy between the FISH and Micro-C data: FISH suggests the formation of several clusters and Micro-C suggests very weak (non-significant) interactions between the centromere of chromosome 8 and those of chromosomes 4 and 5.

Again, what criteria were used to detect enrichment and colocalization in these virtual 4C assays?

As outlined above, we have now implemented an approach to assess the significance of virtual 4C peaks, marking them as significant with a red marker in the relevant figures and detailing how the significance was calculated in the Methods section.

Other peaks could be detected in these 4C analysis. For example, centromere 8 appears to interact with centromere chr10 (Fig S8).

Yes, our virtual 4C data show that the centromeres of chromosomes 8 and 10 cluster. We never say that these interactions do not occur or that interactions between specific centromeres are always exclusive. Below you can find a table showing the average interaction frequency between different centromeres, but we believe that such abstract averages can be misleading and that Figure 7 and Supplementary Figures 9 and 10 are more useful for evaluating interaction frequencies between centromeres.

Figure 3R. We generated a heatmap to visualize pairwise interactions between centromeres. 1) We performed Virtual 4C using each centromere as viewpoint, then we selected bins around centromeres; 2) We calculated the average interaction frequency of the bins grouped by centromere; 3) We averaged for each centromere pair the interaction frequencies of viewpoint and target; 4) We displayed the calculated average frequency in a heatmap.

Did the authors notice any peculiar interaction patterns between centromeres located in core and subtelomeric regions? Do core centromeres only interact with other core centromeres, and vice versa? What is the biological implication of this?

No, we did not observe a specific pattern of interaction between core centromeres and subtelomeric centromeres.

I would place figure S8 in place of Fig 5A, as the FISH validation refers to the interaction data shown in the supplement.

As suggested, we have replaced Figure 5a (now 7a) with Supplementary Figure S8.

For the FISH data, is CIR147 found only on chrs 4, 5, and 8? Why were these particular centromeres chosen?

Yes, CIR147 is only found on chr 4, 5 and 8. The reason we chose this repeat is that we have used FISH probes against it in the past.

The authors mention different classes of repeats in the centromeres, now better resolved with the new sequencing. What are these classes? A table highlighting them would be important.

We now added Supplementary Table 3, listing the different centromere repeat sequences and classes. In addition, we extended the Methods section to describe how the repeats were identified.

5. Regarding the Rabl-like Structure: In my opinion, this section requires more attention. The arguments and experiments leading to this conclusion are not robust. I don't believe the experiments convincingly demonstrate the existence of this type of organization in *T. brucei*.

As outlined in our response to reviewer 1, when we described the genome organization of *T. brucei* as Rabl-like, we had tried to follow the definition of Hoencamp et al.⁷ (Science, 2021, PMID 34045355). As we had interpreted the definition, any genome with observable centromere-centromere or telomere-telomere interactions should be called Rabl-like. In addition, our Hi-C data had revealed a relatively low ratio of intra- to inter-chromosomal interactions, which is reminiscent of other genomes thought to fold into a Rabl-like structure. However, after discussion with other experts in the field of genome architecture, we realized that centromere-centromere interactions are a very common phenomenon in biology, even for genomes of organisms with CT, especially shortly after mitosis⁸ (Solovei and Mirny, Current Opinion in Cell Biology, 2024: PMID: 39180905). As you point out below, we believe that extensive DNA FISH experiments will be necessary to unambiguously describe the organization of the *T. brucei* megabase chromosomes, which may also vary in organization throughout the interphase of the cell cycle. We believe that such experiments would be beyond the scope of this study and have therefore revised the manuscript as also suggested by reviewer 1. We do not state any longer that the *T. brucei* genome assumes a Rabl-like organization.

o Is it possible to detect interactions among multiple centromeres in the micro-C matrices, as seen in yeast?

No, with Micro-C we only pick up interactions between two sites. Multi-valent interactions can only be identified with methods such as SPRITE¹⁷ (Quinodoz et al., Cell, 2018, PMID 29887377) or visualized by FISH.

The authors mention that centromeres are close to the nucleolus, showing FISH images. More images should be shown, along with nucleolar and nuclear membrane markers, simultaneously with telomere labeling, to illustrate the Rab1-like structure where centromeres and telomeres occupy opposite positions in the nucleus.

As mentioned above, we believe that a detailed analysis of the genome architecture by FISH is beyond the scope of this study. We have thus revised the manuscript and don't mention any longer that centromeres localize close to the nucleolus.

Additionally, if centromeres are close to the nucleolus, was it possible to detect interactions between centromeres and rRNA loci by micro-C?

Many rRNA loci are adjacent to centromeres and are therefore part of the interaction network. However, some centromeres also interact with rRNAs that are not adjacent to other centromeres. To make this clear, we have added the annotation of rRNAs to Figure 7.

o The authors claim that *T. brucei* telomeres are clustered, as previously detected by Hi-C and FISH (Muller et al.). However, the Hi-C matrix (Extended Data Fig. 1B) and Micro-C data (from the current article) indicate that telomeres do not cluster. Therefore, *T. brucei* may possess a nuclear structure that differs from both Rab1-like and chromosome territories, independent of telomere clustering.

As detailed above, we agree that the *T. brucei* genome may fold into an organization other than CTs or a Rab1-like structure and have rewritten the manuscript accordingly.

Regarding telomere-telomere interactions, the FISH data of Lowell and Cross¹⁸ (Journal of Cell Science, 2004, PMID 15522895) and Müller et al.⁴ (Nature, 2018, PMID 30333624) and other publications suggest a clustering of telomeres. However, it is difficult (or impossible) to detect an increase in telomere-telomere interactions in genome-wide DNA-DNA interaction frequency plots such as our Hi-C data shown in ED Fig. 1b, Müller et al.⁴) or our Micro-C data shown in Fig. 2 of this manuscript, given the identical sequence of the telomeric repeats. However, for her dissertation¹⁹ (publicly available at <https://opus.bibliothek.uni-wuerzburg.de/frontdoor/index/index/docId/18707>), Dr. Müller used different BESs as a viewpoints and found them to interact more frequently with other subtelomeric regions and with chromosome core regions Fig. 5.1. Thus, we should have cited the dissertation rather than her manuscript when we mentioned that Hi-C data suggest clustering of telomeres. For the revised version of this manuscript, we have removed the statement that Hi-C data suggest telomere clustering.

o Figure S6, which would also be important in suggesting the presence of a Rab1-like structure, is only mentioned in the discussion. However, the results are interesting and seem to conflict with the group's previously published results (Muller et al.), where they show more interaction between subtelomeric than core regions. Please discuss and explore this data.

As addressed above (comment 2), for this study we have used an improved strategy to normalize our Micro-C data and to account for difference in ploidy between chromosome core and subtelomeric regions.

Although our Hi-C and Micro-C maps still show that subtelomeres are more compact than chromosome core regions, the difference in interaction frequency is now less pronounced than in previous heat maps published from our lab. For a quantification of the relative contact probably see below.

Figure 4R. Contact probability as function of genomic separation calculated for subtelomeric regions and for core regions using cooltools²⁰ (Open2C et al., bioRxiv, 2022, doi: 10.1101/2022.10.31.514564).

o I believe the following sentence is incorrect: “Often, organisms with a Rab1-like configuration show a higher frequency of intra-chromosomal interactions compared to organisms whose chromosomes are organized into territories (Supplementary Fig. 12).”

Yes, it should have read "higher frequency of inter-chromosomal interactions". However, the revised manuscript contains an extensively rewritten Discussion that now omits this statement.

Additionally, it needs to be better explained what is meant by "the level of interactions between interchromosomal loci observed in *T. brucei* far exceeds that observed in other eukaryotes." What is the implication of this? Without a conclusion, the sentence feels incomplete.

The revised manuscript contains an extensively rewritten Discussion that now omits this statement.

6. Introduction: The introduction should be reformulated to improve its flow. Some sentences seem disconnected, or their linkage is unclear. For example:

o “Microscopy and Hi-C data from *T. brucei* have suggested that the 3D nuclear organization is important for RNA maturation^{31–33}. Here, highly expressed *T. brucei* genes were found to make frequent inter-chromosomal interactions with a locus that produces the noncoding spliced leader RNA that is trans-spliced to the 5’ end of every mRNA and is thus crucial for mRNA maturation³⁴.” Does "Here" refer to references 31-33 or the current article's data? If it refers to the current article, what are these highly expressed genes? rRNA?

We were referring to the previous studies and have edited the introduction accordingly. Line 86.

o “Here, to investigate the role of inter-chromosomal hubs in the absence of chromosome territories, TADs, and E-P interactions, we generated high-resolution Micro-C contact maps in *T. brucei*.” What is the reference for this? I understand that the lack of CTs has not been shown

before and will be demonstrated in this paper; therefore, it should not be used in the introduction as a justification for using Micro-C.

This is correct, we removed the statement that *T. brucei* lacks CTs.

Minor Comments:

- “In addition, 16 of the 21 annotated centromeres are now fully assembled, ranging in size from 9 to 55 kb” — shouldn’t this be 22 centromeres?

We could only annotate 21 centromeres because there was no enrichment of the KKT2 signal on chromosome 10A, which we used to identify centromeres. In the section about centromere we state that we could only annotate 21 centromeres (line 409).

- Figure 1A and other figures showing the genome (Fig 4, 5, and Supplementary): The colors showing the genomic features and gaps should be better highlighted for easier understanding. The black marking for "telomeric ends" is easily confused with dark blue, so are the centromeres.

We have edited Figures 1, 4, 5 and Supplementary Figures 9 and 10 and hope the features are now easier to recognize.

Reviewer #5 (Remarks to the Author):

This interesting manuscript uses state-of-the-art techniques (Micro-C) to reveal the 3D organization of the *Trypanosoma brucei* genome and show that there is considerable intra- and inter-chromosomal interaction between transcription start sites (TSSs) for the three different classes of RNA polymerase (RNAP). While this has previously been shown for RNAPI (which transcribes rRNA, as well as specific variant surface antigen genes), this now extends the phenomenon to the ~200 polycistronic transcription units (PTUs) of protein coding genes that are unique to this organism (and other Kinetoplastea), as well as (to a lesser extent) the tRNA and snRNA genes transcribed by RNAPIII. This finding highlights the evolutionary significance of inter-chromosomal interaction hubs for RNAP class-specific loci and centromeres and will likely be of considerable interest to the broader scientific community. In general, the manuscript is well-written and concise and is ready for publication with only minor revision, including the following:

1. There appears to be a minor error in Figure 2, where one of the PTUs (at the top of the figure) has arrowheads at both ends. This needs to be corrected.

Yes, we have corrected the arrow in Figure 2.

2. It appears that the Micro-C experiments were carried out on only bloodstream (mammalian) forms of *T. brucei*. While it is probably not critical that they be repeated on procyclic (insect) forms, this should at least be mentioned in the Discussion.

Yes, this is an important point. We now mention in the Discussion that this analysis was performed only with bloodstream form *T. brucei* cells (lines 521-524).

“In this study, we focused on bloodstream form parasites, the stage that infects the mammalian host. Although Hi-C data exist for insect-stage parasites³⁰, no systematic comparison of

genome architecture has been performed for this stage. Thus, it remains to be seen how the genome organization changes during the life cycle of the parasite.”

References

1. Díaz-Viraqué, F., Chiribao, M. L., Libisch, M. G. & Robello, C. Genome-wide chromatin interaction map for *Trypanosoma cruzi*. *Nat. Microbiol.* **8**, 2103–2114 (2023).
2. Beagan, J. A. & Phillips-Cremins, J. E. On the existence and functionality of topologically associating domains. *Nat. Genet.* **52**, 8–16 (2020).
3. Kruse, K., Hug, C. B. & Vaquerizas, J. M. FAN-C: a feature-rich framework for the analysis and visualisation of chromosome conformation capture data. *Genome Biol.* **21**, 303 (2020).
4. Müller, L. S. M. *et al.* Genome organization and DNA accessibility control antigenic variation in trypanosomes. *Nature* **563**, 121–125 (2018).
5. Wolff, J. *et al.* Galaxy HiCExplorer 3: a web server for reproducible Hi-C, capture Hi-C and single-cell Hi-C data analysis, quality control and visualization. *Nucleic Acids Res.* **48**, W177–W184 (2020).
6. Lajoie, B. R., Dekker, J. & Kaplan, N. The Hitchhiker’s guide to Hi-C analysis: Practical guidelines. *Methods* **72**, 65–75 (2015).
7. Hoencamp, C. *et al.* 3D genomics across the tree of life reveals condensin II as a determinant of architecture type. *Science* **372**, 984–989 (2021).
8. Solovei, I. & Mirny, L. Spandrels of the cell nucleus. *Curr. Opin. Cell Biol.* **90**, 102421 (2024).
9. Cordon-Obras, C. *et al.* Identification of sequence-specific promoters driving polycistronic transcription initiation by RNA polymerase II in trypanosomes. *Cell Rep.* **38**, 110221 (2022).
10. Luzak, V. *et al.* SLAM-seq reveals independent contributions of RNA processing and stability to gene expression in African trypanosomes. Preprint at <https://doi.org/10.1101/2024.06.18.599538> (2024).
11. Wedel, C., Förstner, K. U., Derr, R. & Siegel, T. N. GT-rich promoters can drive RNA pol II transcription and deposition of H2A.Z in African trypanosomes. *EMBO J.* **36**, 2581–2594 (2017).
12. Kraus, A. J. *et al.* Distinct roles for H4 and H2A.Z acetylation in RNA transcription in African trypanosomes. *Nat. Commun.* **11**, 1498 (2020).
13. Cosentino, R. O., Brink, B. G. & Siegel, T. N. Allele-specific assembly of a eukaryotic genome corrects apparent frameshifts and reveals a lack of nonsense-mediated mRNA decay. *NAR Genomics Bioinforma.* **3**, lqab082 (2021).
14. Schmidt, M. R., Barcons-Simon, A., Rabuffo, C. & Siegel, T. N. Smoother: on-the-fly processing of interactome data using prefix sums. *Nucleic Acids Res.* **52**, e23–e23 (2024).
15. Roayaei Ardakany, A., Gezer, H. T., Lonardi, S. & Ay, F. Mustache: multi-scale detection of chromatin loops from Hi-C and Micro-C maps using scale-space representation. *Genome Biol.* **21**, 256 (2020).
16. Siegel, T. N. *et al.* Four histone variants mark the boundaries of polycistronic transcription units in *Trypanosoma brucei*. *Genes Dev.* **23**, 1063–1076 (2009).
17. Quinodoz, S. A. *et al.* Higher-Order Inter-chromosomal Hubs Shape 3D Genome Organization in the Nucleus. *Cell* **174**, 744–757.e24 (2018).
18. Lowell, J. E. & Cross, G. A. M. A variant histone H3 is enriched at telomeres in *Trypanosoma brucei*. *J. Cell Sci.* **117**, 5937–5947 (2004).
19. Müller-Hübner, L. The role of nuclear architecture in the context of antigenic variation in *Trypanosoma brucei*. 23534 KB (Universität Würzburg, 2020). doi:10.25972/OPUS-18707.
20. Open2C *et al.* Cooltools: enabling high-resolution Hi-C analysis in Python. *bioRxiv* 2022.10.31.514564 (2022) doi:10.1101/2022.10.31.514564.

Response to reviewers

We are happy to read that we were able to address almost all points raised by reviewers and hope to address the remaining points with this revision.

Reviewer #4 (Remarks to the Author):

The authors have satisfactorily addressed my questions and those of the other reviewers. However, I still have three points that could be clarified.

1. Figure 3a: The new analyses using FAN-C at 10 kb resolution highlight the presence of three types of matrices whose compartmentalization is described as centromere-based, TSS-based, and core x subtelomeric-based. These analyses are indeed interesting; however, for me, in Matrix 2, it is possible to detect both centromere- and TSS-based compartmentalization. Additionally, I believe the matrix 3 does not show greater compaction in the subtelomeric regions but instead appears more compact in the core regions. Finally, in the first matrix, it seems evident that, in addition to centromere-based compartmentalization, TSS-based compartmentalization is also present.

Overall, my impression is that the type of compartmentalization is not exclusive. Does this pattern repeat in other chromosomes? I believe this should be mentioned in the text. To make this clearer, it might also be helpful to indicate in the matrices which of the three types of compartmentalization they correspond to.

We did not mean to imply that compartmentalization is exclusive. In fact, we believe that the opposite is true and that many chromosomes are compartmentalized at different levels at the same time. We have changed the manuscript accordingly.

We now state: "Eigenvectors calculated from the correlation matrix at 10 kb resolution using FAN-C⁸ suggested non-exclusive compartmentalization at three levels."

Since we do not have clear cutoffs on how to define the three compartment types, we would rather not assign them and believe it is more useful to just look at the heat maps or visualize the data in dedicated viewers.

2. The authors use two pipelines (distiller pipeline and mhic) to process and analyze the Micro-C data. However, it is not clear which pipeline was used to generate the input matrix for calculating the insulation score and identifying domains. This is particularly critical since the mhic pipeline leverages multimapped reads, which could impact mappability rates.

In any case, the figure legends where mappability is shown (e.g., Fig. 3 and Supp. 7) should specify the origin of the matrix. Additionally, in Figure S7, it would be important to clarify what is meant by "no visibility biases."

We have edited the legends of Figure 3 and Supplementary Figure 7 to indicate which pipeline we used to calculate the different matrices. We have also added information about the no visibility regions. Details on how these regions were annotated can be found in the Methods section.

3. The "3D clustering of tRNA loci" and "3D interactions between rRNA loci" have also been observed in a closely related species of *T. brucei*. Although these findings are described in a

preprint (<https://www.biorxiv.org/content/10.1101/2024.07.01.601582v1>), both manuscripts share similar conclusions on these topics. This overlap suggests an evolutionary conservation between the two species, making it important to cite the preprint appropriately.

We had cited the preprint before but are now also citing it in the context of clustering of tRNA loci and as another example of domain boundaries enriched in tRNA loci.